# Function Space Particle Optimization for Bayesian Neural Networks

**Ziyu Wang, Tongzheng Ren, Jun Zhu,**[*] **Bo Zhang**
Department of Computer Science & Technology, Institute for Artificial Intelligence,
State Key Lab for Intell. Tech. & Sys., BNRist Center, THBI Lab, Tsinghua University
{wzy196,rtz19970824}@gmail.com, {dcszj,dcszb}@tsinghua.edu.cn

## Abstract

While Bayesian neural networks (BNNs) have drawn increasing attention, their posterior inference remains challenging, due to the high-dimensional and over-parameterized nature. Recently, several highly flexible and scalable variational inference procedures based on the idea of *particle optimization* have been proposed. These methods directly optimize a set of particles to approximate the target posterior. However, their application to BNNs often yields sub-optimal performance, as they have a particular failure mode on over-parameterized models. In this paper, we propose to solve this issue by performing particle optimization directly in the space of regression functions. We demonstrate through extensive experiments that our method successfully overcomes this issue, and outperforms strong baselines in a variety of tasks including prediction, defense against adversarial examples, and reinforcement learning.

## 1 Introduction

Bayesian nerual networks (BNNs) provide a principled approach to reasoning about the *epistemic uncertainty*—uncertainty in model prediction due to the lack of knowledge. Recent work has demonstrated the potential of BNNs in safety-critical applications like medicine and autonomous driving, deep reinforcement learning, and defense against adversarial samples (see e.g. Ghosh et al., 2018; Zhang et al., 2018b; Feng et al., 2018; Smith & Gal, 2018).

Modeling with BNNs involves placing priors on neural network weights, and performing posterior inference with the observed data. However, posterior inference for BNNs is challenging, due to the multi-modal and high dimensional nature of the posterior. Variational inference (VI) is a commonly used technique for practical approximate inference. Traditional VI methods approximate the true posterior with oversimplified distribution families like factorized Gaussians, which can severely limit the approximation quality and induce pathologies such as over-pruning (Trippe & Turner, 2018). These limitations have motivated the recent development of implicit VI methods (Li & Turner, 2018; Shi et al., 2018b), which allow the use of flexible approximate distributions without a tractable density. However, most of the implicit inference methods require to learn a "generator network" that maps a simple distribution to approximate the target posterior. Inclusion of such a generator network can introduce extra complexity, and may become infeasible when the number of parameters is very large, as in the case for BNNs.

Compared with those generator-based methods, particle-optimization-based variational inference (POVI) methods constitute a simpler but more efficient class of implicit VI methods. In an algorithmic perspective, POVI methods iteratively update a set of *particles*, so that the corresponding empirical probability measure approximates the target posterior well. Formally, these methods consider the space of probabilistic measures equipped with different metrics, and simulate a gradient flow that converges to the target distribution. Examples of POVI methods include Stein variational gradient descent (SVGD; Liu & Wang, 2016), gradient flows in the 2-Wasserstein space (Chen et al., 2018), and accelerated first-order methods in the 2-Wasserstein space (Liu et al., 2018).

---

[*]corresponding author

While POVI methods have shown promise in a variety of challenging inference problems, their performance in BNNs is still far from ideal, as with a limited number of particles, it is hard to characterize the highly complex weight-space posterior. The first problem is the curse of dimensionality: Zhuo et al. (2018) and Wang et al. (2018) show that for SVGD with a RBF kernel, particles can collapse to the maximum a posteriori (MAP) estimate as the dimension of parameter increases. One may hope to alleviate such a problem by switching to other POVI methods that could be more suitable for BNN inference; however, this is not the case: BNN is *over-parameterized*, and there exist a large number of local modes in the weight-space posterior that are distant from each other, yet corresponding to the same regression function. Thus a possible particle approximation is to place each particle in a different mode. In prediction, such an approximate posterior will not perform better than a single point estimate. In other words, good approximations for the weight-space posterior do not necessarily perform well in prediction.

To address the above issue, we propose to perform POVI directly for the posterior of regression functions, i.e. the *function-space posterior*, instead for the weight-space posterior. In our algorithm, particles correspond to regression functions. We address the infinite dimensionality of function space, by approximating the function particles by weight-space parameters, and presenting a mini-batch version of particle update. Extensive experiments show that our method avoids the degenerate behavior of weight-space POVI methods, and leads to significant improvements on several tasks, including prediction, model robustness, and exploration in reinforcement learning.

The rest of our paper is organized as follows. We first briefly review BNNs and POVI in Section 2. In Section 3 we present our algorithm for function-space POVI. We compare our method with existing work in Section 4, and finally demonstrate our method's effectiveness in Section 5.

## 2 BACKGROUND

**Bayesian Neural Networks (BNNs)**   Consider a supervised learning task. Let $\mathbf{X} = \{x_i\}_{i=1}^N$ denote the training inputs and $\mathbf{Y} = \{y_i\}_{i=1}^N$ denote the corresponding outputs, with $x_i \in \mathcal{X}$ and $y_i \in \mathcal{Y}$, respectively. Let $f(\cdot; \theta) : \mathcal{X} \to \mathbb{R}^F$ denote a mapping function parameterized by a neural network, where $F$ will be clear according to the task. Then, we can define a conditional distribution $p(y|x, \theta)$ by leveraging the flexibility of function $f(x; \theta)$. For example, for real-valued regression where $\mathcal{Y} = \mathbb{R}$, we could set $F = 1$ and define the conditional distribution as $p(y|x, \theta) = \mathcal{N}(y|f(x; \theta), \sigma^2)$, where $\sigma^2$ is the variance of observation noise; for a classification problem with $K$ classes, we could set $F = K$ and let $p(y|x, \theta) = \mathrm{Multinomial}(y|\mathrm{softmax}(f(x; \theta)))$. A BNN model further defines a prior $p(\theta)$ over the weights $\theta$. Given the training data $(\mathbf{X}, \mathbf{Y})$, one then infers the posterior distribution $p(\theta|\mathbf{X}, \mathbf{Y}) \propto p(\theta)p(\mathbf{Y}|\theta, \mathbf{X})$, and for a test data point $x_{\text{test}}$, $y$ is predicted to have the distribution $p(y_{\text{test}}|x_{\text{test}}, \mathbf{X}, \mathbf{Y}) = \int p(y_{\text{test}}|x_{\text{test}}, \theta)p(d\theta|\mathbf{X}, \mathbf{Y})$.

Posterior inference for BNNs is generally difficult due to the high-dimensionality of $\theta$. The *over-parameterized* nature of BNNs further exacerbates the problem: for over-parameterized models, there exist multiple $\theta$ that correspond to the same likelihood function $p(y|x, \theta)$. One could easily obtain an exponential number of such $\theta$, by reordering the weights in the network. Each of the $\theta$ can be a mode of the posterior, which makes approximate inference particularly challenging for BNNs.

**Particle-Optimization based Variational Inference (POVI)**   Variational inference aims to find an approximation of the true posterior. POVI methods (Liu & Wang, 2016; Chen et al., 2018) view the approximate inference task as minimizing some energy functionals over probability measures, which obtain their minimum at the true posterior. The optimization problem is then solved by simulating a corresponding gradient flow in certain metric spaces, i.e. to simulate a PDE of the form

$$\partial_t q_t = -\nabla \cdot (\mathbf{v} \cdot q_t),$$

where $q_t$ is the approximate posterior at time $t$, and $\mathbf{v}$ is the gradient flow depending on the choice of metric and energy functional. As $q_t$ can be arbitrarily flexible, it cannot be maintained exactly in simulation. Instead, POVI methods approximate it with a set of particles $\{\theta^{(i)}\}_{i=1}^n$, i.e. $q_t(\theta) \approx \frac{1}{n} \sum_{i=1}^n \delta(\theta - \theta_t^{(i)})$, and simulate the gradient flow with a discretized version of the ODE $d\theta_t^{(i)}/dt = -\mathbf{v}(\theta_t^{(i)})$. In other words, in each iteration, we update the particles with

$$\theta_{\ell+1}^i \leftarrow \theta_\ell^i - \epsilon_\ell \mathbf{v}(\theta_\ell^i), \tag{1}$$

Table 1: Common choices of the gradient flow $\mathbf{v}$ in POVI methods, where $k$ denotes a kernel, and $\mathbf{K}_{ij} := k(\theta^{(i)}, \theta^{(j)})$ is the gram matrix. We omit the subscript $\ell$ for brevity.

| Method | $-\mathbf{v}(\theta^{(i)})$ |
|---|---|
| SVGD (Liu & Wang, 2016) | $\frac{1}{n} \sum_{j=1}^{n} \mathbf{K}_{ij} \nabla_{\theta^{(j)}} \log p(\theta^{(j)}|\mathbf{x}) + \nabla_{\theta^{(j)}} \mathbf{K}_{ij}$ |
| w-SGLD-B (Chen et al., 2018) | $\nabla_{\theta^{(i)}} \log p(\theta^{(i)}|\mathbf{x}) + \sum_{j=1}^{n} \nabla_{\theta^{(j)}} \mathbf{K}_{ji} / \sum_{k=1}^{n} \mathbf{K}_{jk}$ $+ \sum_{j=1}^{n} \nabla_{\theta^{(j)}} \mathbf{K}_{ji} / \sum_{k=1}^{n} \mathbf{K}_{ik}$ |
| pi-SGLD (Chen et al., 2018) | sum of $-\mathbf{v}$ in SVGD and w-SGLD-B |
| GFSF (Liu et al., 2018) | $\nabla_{\theta^{(i)}} \log p(\theta^{(i)}|\mathbf{x}) + \sum_{j=1}^{n} (\mathbf{K}^{-1})_{ij} \nabla_{\theta^{(j)}} \mathbf{K}_{ij}$ |

where $\epsilon_\ell$ is the step-size at the $\ell$-th iteration. Table 1 summarizes the common choices of the gradient flow $\mathbf{v}$ for various POVI methods. We can see that in all cases, $\mathbf{v}$ consists of a (possibly smoothed) **log posterior gradient term**, which pushes particles towards high-density regions in the posterior; and a **repulsive force term** (e.g. $\nabla_{\theta^{(j)}} \mathbf{K}_{ij}$ for SVGD), which prevents particles from collapsing into a single MAP estimate.

While the flexibility of POVI methods is unlimited in theory, the use of finite particles can make them un-robust in practice, especially when applied to high-dimensional and over-parameterized models. The problem of high dimensionality is investigated in Zhuo et al. (2018) and Wang et al. (2018). Here we give an intuitive explanation of the over-parameterization problem[1]: in an over-parameterized model like BNN, the target posterior has a large number of modes that are sufficiently distant from each other, yet corresponding to the same regression function $f$. A possible convergence point for POVI with finite particles is thus to occupy all these modes, as such a configuration has a small 2-Wasserstein distance (Ambrosio et al., 2008) to the true posterior. In this case, prediction using these particles will not improve over using the MAP estimate. Such a degeneracy is actually observed in practice; see Section 5.1 and Appendix A.1.

## 3 FUNCTION SPACE PARTICLE OPTIMIZATION

To address the above degeneracy issue of existing POVI methods, we present a new perspective as well as a simple yet efficient algorithm to perform posterior inference in the space of regression functions, rather than in the space of weights.

Our method is built on the insight that when we model with BNNs, there exists a map from the network weights $\theta$ to a corresponding regression function $f$, $\theta \mapsto f(\cdot; \theta)$, and the prior on $\theta$ implicitly defines a prior measure on the space of $f$, denoted as $p(f)$. Furthermore, the conditional distribution $p(y|x, \theta)$ also corresponds to a conditional distribution of $p(y|x, f)$. Therefore, posterior inference for network weights can be viewed as posterior inference for the regression function $f$.

A nice property of the function space inference is that it does not suffer from the over-parameterization problem. However, it is hard to implement, as the function space is infinite-dimensional, and the prior on it is implicitly defined. We will present a simple yet effective solution to this problem. In the sequel, we will use boldfaced symbols $\mathbf{x}$ (or $\mathbf{y}$) to denote a subset of samples from $\mathcal{X}$ (or $\mathcal{Y}$)[2]. We denote the approximate posterior measure as $q(f)$. For any finite subset $\mathbf{x}$, we use $f(\mathbf{x})$ to denote the vector-valued evaluations of the regression function on $\mathbf{x}$, and define $p(f(\mathbf{x})), q(f(\mathbf{x}))$ accordingly. We will use the notation $[n] := \{1, 2, \ldots, n\}$.

### 3.1 FUNCTION SPACE PARTICLE OPTIMIZATION ON A FINITE INPUT SPACE

For the clarity of presentation, we start with a simple setting, where $\mathcal{X}$ is a finite set and the gradient for the (log) *function-space prior*, $\nabla_{f(\mathbf{x})} \log p(f(\mathbf{x}))$ is available for any $\mathbf{x}$. These assumptions will be relaxed in the subsequent section. In this case, we can treat the function values $f(\mathcal{X})$ as the parameter to be inferred, and apply POVI in this space. Namely, we maintain $n$ particles

---

[1] People familiar with SVGD could argue that this issue can be mitigated by choosing a reasonable kernel function in $\mathbf{v}$, e.g. a kernel defined on $f(\cdot; \theta)$. We remark that similar kernels do not work in practice. We provide detailed experiments and discussion in Appendix B.

[2] Note $\mathbf{x}$ does not need to be a subset of $\mathbf{X}$.

$f^1(\mathcal{X}), \ldots, f^n(\mathcal{X})$; and in step $\ell$, update each particle with

$$f^i_{\ell+1}(\mathcal{X}) \leftarrow f^i_\ell(\mathcal{X}) - \epsilon_\ell \mathbf{v}[f^i_\ell(\mathcal{X})]. \tag{2}$$

This algorithm is sound when $f(\mathcal{X})$ is finite-dimensional, as theories on the consistency of POVI (e.g. Liu, 2017) directly apply. For this reason, we refer to this algorithm as the *exact version* of function-space POVI, even though the posterior is still approximated by particles.

However, even in the finite-$\mathcal{X}$ case, this algorithm can be inefficient for large $\mathcal{X}$. We address this issue by approximating function-space particles in a weight space, and presenting a mini-batch version of the update rule. As we shall see, these techniques naturally generalize to the case when $\mathcal{X}$ is infinite.

### 3.1.1 Parametric Approximation to Particle Functions

Instead of explicitly maintaining $f(x)$ for all $x \in \mathcal{X}$, we can represent a function $f$ by a parameterized neural network. Although any flexible network can be used, here we choose the original network with parameters $\theta$ of the BNN model, which can faithfully represent any function in the support of the function prior. Note that although we now turn to deal with weights $\theta$, our method is significantly different from the existing POVI methods, as explained below in Remark 3.2 and Appendix D. Formally, our method maintains $n$ weight-space particles $\theta^i_\ell$ ($i \in [n]$) at iteration $\ell$, and defines the update rule as follows:

$$\theta^i_{\ell+1} \leftarrow \theta^i_\ell - \epsilon_\ell \left( \frac{\partial f(\mathcal{X}; \theta^i_\ell)}{\partial \theta^i_\ell} \right)^\top \mathbf{v}[f^i_\ell(\mathcal{X})], \tag{3}$$

where we use the shorthand $f^i_\ell(\cdot) := f(\cdot; \theta^i_\ell)$. As the weights correspond to $n$ regression functions, the rule (3) essentially updates the particles of $f$. In the following remarks, we relate (3) to the "exact rule" (2), and discuss its implications.

**Remark 3.1.** *((3) **as a single step of GD**) The update rule (3) essentially is a one-step gradient descent (GD) to minimize the squared distance between $f^i_{\ell+1}(\mathcal{X})$ and $f^i_\ell(\mathcal{X}) - \epsilon_\ell \mathbf{v}[f^i_\ell(\mathcal{X})]$ (the exact function-space update (2)) under the parametric representation of $f$. Similar strategies have been successfully used in deep reinforcement learning (Mnih et al., 2015).*
*Also note that (3) is easy to implement, as it closely relates to the familiar back-propagation (BP) procedure for the maximum likelihood estimation (MLE). Namely, the GD for MLE corresponds to the update $\theta_{\ell+1} \leftarrow \theta_\ell - \epsilon_\ell \left( \frac{\partial f(\mathcal{X}; \theta_\ell)}{\partial \theta_\ell} \right) \nabla_{f(\mathcal{X}; \theta_\ell)} \log p(\mathbf{Y}|\mathbf{X}, f(\mathcal{X}; \theta_\ell))$, where $\nabla_{f(\mathcal{X}; \theta_\ell)} \log p(\mathbf{Y}|\mathbf{X}, f(\mathcal{X}; \theta_\ell))$ is commonly referred to as the "error signal of top-layer network activation" in BP. In our algorithm, this term is replaced with $\mathbf{v}$. Recall that $\mathbf{v}$ in commonly used POVI algorithms is the sum of a possibly smoothed log posterior gradient, which is similar to $\nabla_f \log p(\mathbf{Y}|\mathbf{X}, f)$ used in MLE training, and the repulsive force (RF) term. Thus our algorithm can be seen as BP with a modified top-layer error signal.*

**Remark 3.2.** *(**Relation between** (3), **ensemble training, and weight-space POVI**) The widely used ensemble training method (Opitz & Maclin, 1999) obtains $n$ MAP estimates separately via GD. As stated above, our algorithm can be seen as a BP procedure with a modified top-layer error signal, thus it is closely related to ensemble training. The main difference is that our algorithm adds a RF term to the error signal, which pushes the prediction of each particle away from that of others. As an example, consider function-space SVGD with RBF kernels. The function-space RF term is $\frac{1}{n} \sum_j \nabla_{f^j_\ell} k(f^i_\ell, f^j_\ell) \propto \sum_{j \neq i} (f^i_\ell - f^j_\ell) k(f^i_\ell, f^j_\ell)$ (see Appendix D), which drives $f^i_\ell$ away from $f^j_\ell$. Our algorithm thus enhances ensemble training, in which the predictions of all particles converge to the MAP and could suffer from overfitting.*
*The relation to ensemble training also exists in weight-space POVI methods (Liu & Wang, 2016). However, the RF in those methods is determined by a weight-space kernel. As discussed in Section 2, commonly used weight-space kernels cannot characterize the distance between model predictions in over-parameterized models like BNNs. In contrary, the function-space RF in our method directly accounts for the difference between model predictions, and is far more efficient. We will present empirical evidence in Section 5.1. Derivations supporting this remark are included in Appendix D.*

**Remark 3.3.** *Finally, the update rule (2) corresponds to a gradient flow in a Wasserstein space of measures of functions, while it is not clear if similar results exist for rule (3). As (3) is simple to implement, and has the desirable properties discussed in Remark 3.2, we consider (3) as a reasonable*

*approximation to the exact gradient-flow based algorithm. As we show empirically in Appendix C, it does not impact convergence; a thorough theoretical treatment is left for future work.*

### 3.1.2 MINI-BATCH VERSION OF THE PARTICLE UPDATE

One shortcoming of the above procedure is that it still needs to iterate over the whole set $\mathcal{X}$ when calculating the update rule (3), which can be inefficient for large $\mathcal{X}$. We further improve the efficiency by presenting a mini-batch version, i.e., in each iteration, we draw a mini-batch with $B$ elements, $\mathbf{x} \sim \mu$, for an *arbitrary* distribution $\mu$ supported on $\mathcal{X}^B$; we then replace the update rule (3) with evaluations on $\mathbf{x}$, i.e.

$$\theta_{\ell+1}^i \leftarrow \theta_\ell^i - \epsilon_\ell \left( \frac{\partial f(\mathbf{x}; \theta_\ell^i)}{\partial \theta_\ell^i} \right)^\top \mathbf{v}[f_\ell^i(\mathbf{x})]. \tag{4}$$

**Justification of** (4)   As above, we have related the full-input-space update rule (3) to (2), a gradient flow (GF) in the Wasserstein space of measures of functions (Liu et al., 2017; Chen et al., 2018; Liu et al., 2018). Similarly, for any fixed $\mathbf{x}$, the update (4) corresponds to a GF in the same space, denoted as $\partial_t q = -\nabla \cdot (\mathbf{v_x} \cdot q_t)$. Simulating (4) with $\mathbf{x}$ sampled from $\mu$ can be seen as a stochastic approximation to simulating the *"averaged GF"*, $\partial_t q = -\nabla \cdot \mathbb{E}_{\mathbf{x} \sim \mu}(\mathbf{v_x} \cdot q_t)$. It is known that under certain assumptions, stochastic approximation to GF simulation does not impact convergence[3], so it suffices to justify the simulation of the "averaged GF". For any fixed and finite $\mathbf{x}$, it is known that the GF $\partial_t q = -\nabla \cdot (\mathbf{v_x} \cdot q_t)$ minimizes a geodesically convex energy, denoted as $\mathcal{E}_{\mathbf{x}}[q(f)]$, whose minimizer $q(f)$ must satisfy $q(f(\mathbf{x})) = p(f(\mathbf{x})|\mathbf{X}, \mathbf{Y})$ (see Chen et al. (2018) for definition of $\mathcal{E}_{\mathbf{x}}$). Thus the averaged GF minimizes the *averaged energy* $\mathbb{E}_{\mathbf{x} \sim \mu} \mathcal{E}_{\mathbf{x}}[q(f)]$ (Ambrosio et al., 2008). We discuss the implication of using $\mathbb{E}_{\mathbf{x} \sim \mu} \mathcal{E}_{\mathbf{x}}[q(f)]$ as the variational objective below.

If the posterior process can be uniquely determined by almost all[4] $B$-dimensional marginals $\{p(f(\mathbf{x})|\mathbf{X}, \mathbf{Y}) : \mathbf{x} \in \text{supp}(\mu)\}$, it will clearly become the unique minimizer of the averaged energy, and simulation of the averaged gradient flow yields the true posterior. As a concrete example, if the posterior is a Gaussian process, a sufficient condition is to set $B \geq 2$, and $\mu$ to be the product measure of measures with support $\mathcal{X}$, and measures with support $\mathbf{X}$.[5]

Even if not all marginals of the variational posterior $q(f)$ converge to those of the true posterior, the averaged energy serves as a good variational objective on its own. This is because it measures the average approximation error of the lower-dimensional marginals of the posterior, weighted by $\mu$; and in most applications, we are only interested in this quantity. For example, in supervised learning tasks with i.i.d. training and test samples, we need to minimize the average approximation error of posterior predictive mean and variance. In this case, it is sufficient to obtain a good approximation of all two-dimensional marginal distributions, weighted by the input distribution; and we can set the full-support component in $\mu$ to be the kernel density estimation (KDE) of the training set.

We reiterate that the above condition on $\mu$ does not imply $\mu$ must be "uniform" on $\mathcal{X}$ in any sense, or each of its $B$ marginals must be identically distributed.

**Computation of** (4)   To implement (4) we need to compute $\mathbf{v}[f_\ell^i(\mathbf{x})]$. As shown in Table 1, it requires the specification of a kernel and access to (the gradient of) the log posterior density, both on the $B$-dimensional space spanned by $f(\mathbf{x})$; it also requires the specification of $\mu$. For kernels, any positive definite kernels can be used. In our experiments, we choose the RBF kernel with the median heuristic for bandwidth, as is standard in POVI implementations (Liu & Wang, 2016).

The log posterior gradient consists of the gradient of log prior and that of log likelihood. As in this subsection, we assume $\nabla_{\mathbf{x}} \log p(f(\mathbf{x}))$ is known, we only consider the log likelihood. As is standard

---

[3]Simulation of gradient flows corresponds to optimization in the same space, so results about stochastic optimization on Riemannian manifold apply (e.g. Theorem 1 in Ombach & Tarłowski (2012)). Stochastic optimization on Riemannian manifolds is not as well-studied as their Euclidean counterparts, and assumptions introduced in existing work do not apply in our case (e.g. many results requires compactness (Ombach & Tarłowski, 2012; Bonnabel et al., 2013; Tripuraneni et al., 2018)). However, they should serve as adequate motivations of our method.

[4]The concise condition is that for any $\mathcal{X}_s \subset \text{supp}(\mu)$ s.t. $\mu(\mathcal{X}_s) = 1$, $\{p(f(\mathbf{x})|\mathbf{X}, \mathbf{Y}) : \mathbf{x} \in \mathcal{X}_s\}$ uniquely determines the posterior process.

[5]Recall $\mu$ is a measure on $\mathcal{X}^B$, i.e. a sample from $\mu$ contain $B$ points in the input space.

in large-scale inference, we approximate it using mini-batches, i.e. to approximate it with (a scaled version of) $\log p(\mathbf{y}_b|\mathbf{x}_b, f_\ell^i(\mathbf{x}_b))$, where $(\mathbf{x}_b, \mathbf{y}_b)$ is a mini-batch of the training set. The requirement to sample $(\mathbf{x}_b, \mathbf{y}_b)$ is implemented by specifying an appropriate form of $\mu$: we define $\mu$ in such a way that a sample from $\mu$ consists of $B' < B$ samples $\mathbf{x}_b$ from the training set, and $B - B'$ i.i.d. samples from a continuous distribution $\nu$ over $\mathcal{X}$. This is a valid choice, as stated before. Now we can use the training-set part of samples to compute the log likelihood. Finally, the continuous component $\nu$ can be chosen as the KDE of $\mathbf{X}$, when the test set is identically distributed as the training set, or incorporate distributional assumptions of the test set otherwise. For example, for unsupervised domain adaptation (Ganin & Lempitsky, 2015), we can use the KDE of the unlabeled test-domain samples.

Summing up, we present a simple yet efficient function-space POVI procedure, as outlined in Algorithm 1. As we will show empirically in Appendix C, our algorithm converges robustly in practice.

---

**Algorithm 1** Function Space POVI for Bayesian Neural Network

---

1: **Input:** (Possibly approximated) function-space prior $p(f(\mathbf{x}))$ for any finite $\mathbf{x}$; training set $(\mathbf{X}, \mathbf{Y})$; a continuous distribution $\nu$ supported on $\mathcal{X}$ (e.g. the KDE of $\mathbf{X}$); a choice of $\mathbf{v}$ from Table 1; batch size $B, B'$; and a set of initial particles $\{\theta_0^i\}_{i=1}^n$.
2: **Output:** A set of particles $\{\theta^i\}_{i=1}^n$, such that $f(\cdot; \theta^i)$ approximates the target distribution.
3: **for** iteration $\ell$ **do**
4:     Sample a mini-batch $\mathbf{x}_b, \mathbf{y}_b$ from the training set, and $\tilde{x}_{1 \ldots B - B'} \overset{i.i.d.}{\sim} \nu$. Denote $\mathbf{x} = \mathbf{x}_b \cup \{\tilde{x}_i : i \in [B - B']\}$.
5:     For each $i \in [n]$, calculate the function space POVI update $\mathbf{v}[f_\ell^i(\mathbf{x})]$.
6:     For each $i \in [n]$, calculate $\theta_{\ell+1}^i$ according to (4).
7:     Set $\ell \leftarrow \ell + 1$.
8: **end for**

---

## 3.2 GENERAL SCENARIOS

We now relax the assumptions in Section 3.1 to make our setting more practical in real applications. Below we address the infinity of $\mathcal{X}$ and the lack of function-space prior gradient in turn.

**Infinite Set $\mathcal{X}$**    While we assume $\mathcal{X}$ is a finite set to make Section 3.1.1 more easily understood, our algorithm works no matter $\mathcal{X}$ is finite or not: as our algorithm works with mini-batches, when $\mathcal{X}$ is infinite, we can also sample $\mathbf{x}$ from $\mathcal{X}$ and apply the whole procedure.

**Function-Space Prior Gradient**    The function space prior for BNN is implicitly defined, and we do not have access to its exact gradient. While we could in principle utilize gradient estimators for implicit models (Li & Turner, 2018; Shi et al., 2018a), we opt to use a more scalable workaround in implementation, which is to approximate the prior measure with a Gaussian process (GP). More specifically, given input $\mathbf{x}$, we draw samples $\tilde{\theta}^1, \ldots, \tilde{\theta}^k$ from $p(\theta)$ and construct a multivariate normal distribution that matches the first two moments of $\tilde{p}(f(\mathbf{x})) = \frac{1}{k} \sum_{j=1}^k \delta(f(\mathbf{x}) - f(\mathbf{x}; \tilde{\theta}^j))$. We expect this approximation to be accurate for BNNs, because under assumptions like Gaussian prior on weights, as each layer becomes infinitely wide, the prior measure determined by BNNs will converge to a GP with a composite kernel (de G. Matthews et al., 2018; Garriga-Alonso et al., 2018; Lee et al., 2018; Novak et al., 2019).

A small batch size is needed to reduce the sample size $k$, as otherwise the covariance estimate in GP will have a high variance. While our procedure works for fairly small $B$ (e.g. $B \geq 2$ for a GP posterior), we choose to use separate batches of samples to estimate the gradient of the log prior and log likelihood. In this way, a much larger batch size could be used for the log likelihood estimate.

## 4 RELATED WORK

Our algorithm addresses the problem of over-parameterization, or equivalently, non-identifiability[6]. A classical idea addressing non-identifiability is to introduce alternative metrics in the weight space,

---

[6]Non-identifiability means there are multiple parameters that correspond to the same model. *Local non-identifiability* means for any neighborhood $U$ of parameter $\theta$, there exists $\theta' \in U$ s.t. $\theta'$ and $\theta$ represent the

so that parameters corresponding to similar statistical models are closer under that metric. The typical choice is the Fisher information metric, which has been utilized to improve Markov Chain Monte Carlo methods (Girolami & Calderhead, 2011), variational inference (Zhang et al., 2018a) and gradient descent (Amari, 1997). While such methods explore locally non-identifiable parameter regions more rapidly than their weight-space counterparts (Amari, 2016), they still suffer from global non-identifiability, which frequently occurs in models like Bayesian neural networks. Our work takes one step further: by defining metrics in the function space, we address local and global non-identifiability simultaneously.

Closely related to our work is the variational implicit process (VIP; Ma et al., 2018), which shares the idea of function space inference. VIP addresses inference and model learning simultaneously; however, their inference procedure did not address the challenge of inference in complex models: the inference algorithm in VIP draws $S$ prior functions from $p(f)$, and fits a Bayesian linear regression model using these functions as features. As $S$ is limited by the computational budget, such an approximation family will have problem scaling to more complex models. We present comparisons to VIP in Appendix A.2.3.

Approximate inference for BNN is a rich field. Under the VI framework, apart from the implicit VI methods mentioned in Section 1, Louizos & Welling (2017) proposed a hierarchical variational model, which approximates $p(\theta|\mathbf{x}, \mathbf{y})$ with $q(\theta) = \int q(\theta|\mathbf{z})q(\mathbf{z})d\mathbf{z}$, where $\mathbf{z}$ represents layer-wise multiplicative noise, parameterized by normalizing flows. While this approach improves upon plain single-level variational models, its flexibility is limited by a oversimplified choice of $q(\theta|\mathbf{z})$. Such a trade-off between approximation quality and computational efficiency is inevitable for weight-space VI procedures. Another line of work use stochastic gradient Markov Chain Monte Carlo (SG-MCMC) for approximate inference (Li et al., 2016; Chen et al., 2014). While SG-MCMC converges to the true posterior asymptotically, within finite time it produces correlated samples, and has been shown to be less particle-efficient than the deterministic POVI procedures (Liu & Wang, 2016; Chen et al., 2018). Finally, there are other computationally efficient approaches to uncertainty estimation, e.g. Monte-Carlo dropout (Gal & Ghahramani, 2016), batch normalization (Hron et al., 2018), and efficient implementations of factorized Gaussian approximation (e.g., Blundell et al., 2015; Zhang et al., 2018a; Khan et al., 2018).

## 5 EVALUATION

In this section, we evaluate our method on a variety of tasks. First, we present a qualitative evaluation on a synthetic regression dataset. We then evaluate the predictive performance on several standard regression and classification datasets. Finally, we assess the uncertainty quality of our method on two tasks: defense against adversarial attacks, and contextual bandits.

We compare with strong baselines. For our method, we only present results implemented with SVGD for brevity (abbreviated as "f-SVGD"); results using other POVI methods are similar, and can be found in Appendix A.1 and A.2.2. Unless otherwise stated, baseline results are directly taken from the original papers, and comparisons are carried out under the same settings.

Code for the experiments will be available at `https://github.com/thu-ml/fpovi`. The implementation is based on ZhuSuan (Shi et al., 2017).

### 5.1 SYNTHETIC DATA AND THE OVER-PARAMETERIZATION PROBLEM

To evaluate the approximation quality of our method qualitatively, and to demonstrate the curse-of-dimensionality problem encountered by weight space POVI methods, we first experiment on a simulated dataset. We follow the simulation setup in Sun et al. (2017): for input, we randomly generate 12 data points from Uniform$(0, 0.6)$ and 8 from Uniform$(0.8, 1)$. The output $y_n$ for input $x_n$ is modeled as $y_n = x_n + \epsilon_n + \sin(4(x_n + \epsilon_n)) + \sin(13(x_n + \epsilon_n))$, where $\epsilon_n \sim \mathcal{N}(0, 0.0009)$. The model is a feed-forward network with 2 hidden layers and ReLU activation; each hidden layer has 50 units. We use 50 particles for weight space SVGD and our method, and use Hamiltonian

---

same model, i.e., they correspond to the same likelihood function. *Global non-identifiability* means there exists such $\theta'$, but not necessarily in all neighborhoods.

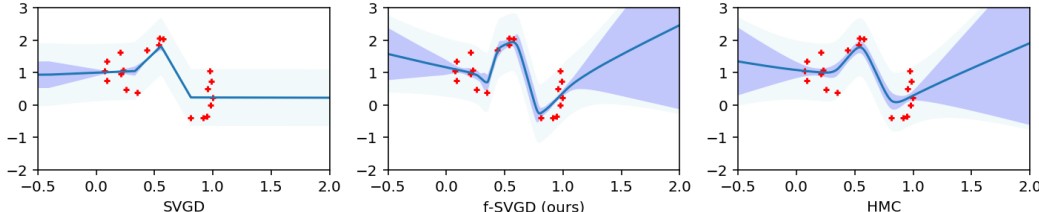

Figure 1: Approximate posterior obtained by different methods. Dots indicate observations, solid line indicates predicted mean, light shaded area corresponds to the predictive credible interval, and dark shaded area corresponds to the credible interval for mean estimate.

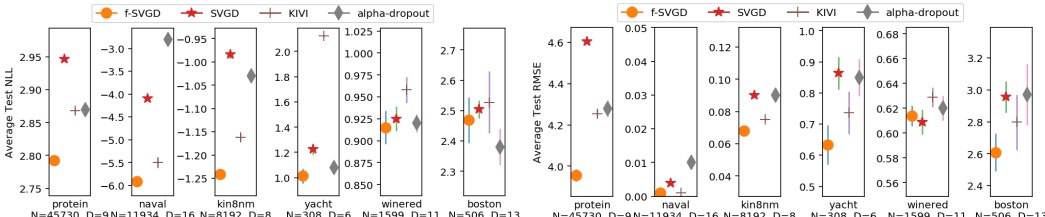

Figure 2: Average test RMSE and predictive negative log-likelihood, on UCI regression datasets. Smaller (lower) is better. Best viewed in color.

Monte Carlo (HMC) to approximate the ground truth posterior. We plot $95\%$ credible intervals for prediction and mean estimate, representing epistemic and aleatoric[7] uncertainties respectively.

Fig. 1 shows the results. We can see our method provides a reasonable approximation for epistemic uncertainty, roughly consistent with HMC; on the other hand, weight-space POVI methods severely underestimate uncertainty. Furthermore, we found that such pathology exists in all weight-space POVI methods, and amplifies as model complexity increases; eventually, all weight-space methods yield degenerated posteriors concentrating on a single function. We thus conjecture it is caused by the over-parameterization problem in weight space. See Appendix A.1 for related experiments.

## 5.2 PREDICTIVE PERFORMANCE

Following previous work on Bayesian neural networks (e.g. Hernández-Lobato & Adams, 2015), we evaluate the predictive performance of our method on two sets of real-world datasets: a number of UCI datasets for real-valued regression, and the MNIST dataset for classification.

### 5.2.1 UCI REGRESSION DATASET

On the UCI datasets, our experiment setup is close to Hernández-Lobato & Adams (2015). The model is a single-layer neural network with ReLU activation and 50 hidden units, except for a larger dataset, Protein, in which we use 100 units. The only difference to Hernández-Lobato & Adams (2015) is that we impose an inverse-Gamma prior on the observation noise, which is also used in e.g. Shi et al. (2018b) and Liu & Wang (2016). Detailed experiment setup are included in Appendix A.2.1, and full data for our method in Appendix A.2.2.

We compare with the original weight-space Stein variational gradient descent (SVGD), and two strong baselines in BNN inference: kernel implicit variational inference (KIVI, Shi et al., 2018b), and variational dropout with $\alpha$-divergences (Li & Gal, 2017). The results are summarized in Fig. 2. We can see that our method has superior performance in almost all datasets.

In addition, we compare with another two state-of-the-art methods for BNN inference: multiplicative normalizing flows and Monte-Carlo batch normalization. As the experiment setup is slightly different following (Azizpour et al., 2018), we report the results in Appendix A.2.3. In most cases, our method also compares favorably to these baselines.

---

[7]predictive uncertainty due to the noise in the data generating process.

Table 2: Test error on the MNIST dataset. **Boldface** indicates the best result.

| Method | BBB (Gaussian Prior) | BBB (Scale Mixture Prior) | KIVI | f-SVGD |
|---|---|---|---|---|
| **Test Error** | 1.82% | 1.36% | 1.29% | **1.21%** |

### 5.2.2 MNIST CLASSIFICATION DATASET

Following previous work such as Blundell et al. (2015), we report results on the MNIST handwriting digit dataset.We use a feed-forward network with two hidden layers, 400 units in each layer, and ReLU activation, and place a standard normal prior on the network weights. We choose this setting so the results are comparable with previous work.

We compare our results with vanilla SGD, Bayes-by-Backprop (Blundell et al. (2015)), and KIVI. For our method, we use a mini-batch size of 100, learning rate of $2 \times 10^{-4}$ and train for 1,000 epochs. We hold out the last 10,000 examples in training set for model selection. The results are summarized in Table 2. We can see that our method outperform all baselines.

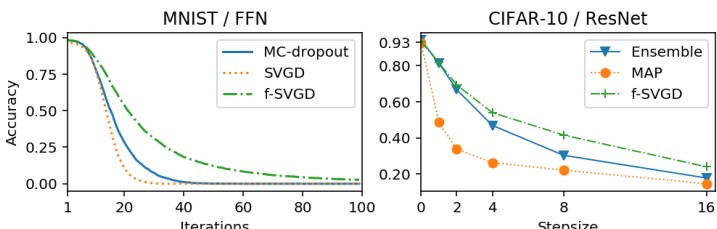

Figure 3: Accuracy on adversarial examples.

### 5.3 ROBUSTNESS AGAINST ADVERSARIAL EXAMPLES

Deep networks are vulnerable to adversarial noise, with many efficient algorithms to craft such noise (cf. e.g. Dong et al., 2018), while defending against such noise is till a challenge (e.g. Pang et al., 2018b;a). It is hypothesized that Bayesian models are more robust against adversarial examples due to their ability to handle epistemic uncertainty (Rawat et al., 2017; Smith & Gal, 2018). This hypothesis is supported by Li & Gal (2017), in a relatively easier setup with feed-forward networks on MNIST; to our knowledge, no results are reported using more flexible approximate inference techniques. In this section, we evaluate the robustness of our method on a setting compatible to previous work, as well as a more realistic setting with ResNet-32 on the CIFAR-10 dataset. We briefly introduce the experiment setup here; detailed settings are included in Appendix A.3.

On the MNIST dataset, we follow the setup in Li & Gal (2017), and experiment with a feed-forward network. We use the iterative fast gradient sign method (I-FGSM) to construct targeted white-box attack samples. In each iteration, we limit the $\ell^\infty$ norm of the perturbation to 0.01 (pixel values are normalized to the range of $[0, 1]$). We compare our method with vanilla SVGD, and MC Dropout.

On the CIFAR-10 dataset, we use the ResNet-32 architecture (He et al., 2016a). As dropout requires modification of the model architecture, we only compare with the single MAP estimate, and an ensemble model. We use 8 particles for our method and the ensemble baseline. We use the FGSM method to construct white-box untargeted attack samples.

Fig. 3 shows the results for both experiments. We can see our method improves robustness significantly, both when compared to previous approximate BNN models, and baselines in the more realistic setting.

### 5.4 IMPROVED EXPLORATION IN CONTEXTUAL BANDIT

Finally, we evaluate the approximation quality of our method on several contextual bandit problems, Contextual bandit is a standard reinforcement learning problem. It is an arguably harder task than supervised learning for BNN approximation methods, as it requires the agent to balance between exploitation and exploration, and decisions based on poorly estimated uncertainty will lead to catastrophic performance through a feedback loop (Riquelme et al., 2018). Problem background and experiment details are presented in Appendix A.4.

We consider the Thompson sampling algorithm with Bayesian neural networks. We use a feed-forward network with 2 hidden layers and 100 ReLU units in each layer. Baselines include other approximate inference methods including Bayes-by-Backprop and vanilla SVGD, as well as other uncertainty estimation procedures including Gaussian process and frequentist bootstrap. We use the mushroom and wheel bandits from Riquelme et al. (2018).

The cumulative regret is summarized in Table 3. We can see that our method provides competitive performance compared to the baselines, and outperforming all baselines by a large margin in the wheel bandit, in which high-quality uncertainty estimate is especially needed.

Table 3: Cumulative regret in different bandits. Results are averaged over 10 trials.

|  | BBB | GP | Bootstrap | f-SVGD |
|---|---|---|---|---|
| Mushroom | $19.15 \pm 5.98$ | $16.75 \pm 1.63$ | $\mathbf{2.71 \pm 0.22}$ | $4.39 \pm 0.39$ |
| Wheel | $55.77 \pm 8.29$ | $60.80 \pm 4.40$ | $42.16 \pm 7.80$ | $\mathbf{7.54 \pm 0.41}$ |

## 6 CONCLUSION

We present a flexible approximate inference method for Bayesian regression models, building upon particle-optimization based variational inference procedures. The newly proposed method performs POVI on function spaces, which is scalable and easy to implement and overcomes the degeneracy problem in direct applications of POVI procedures. Extensive experiments demonstrate the effectiveness of our proposal.

## ACKNOWLEDGEMENTS

The work was supported by the National NSF of China (Nos. 61621136008, 61620106010), the National Key Research and Development Program of China (No. 2017YFA0700900), Beijing Natural Science Foundation (No. L172037), Tsinghua Tiangong Institute for Intelligent Computing, the NVIDIA NVAIL Program, a project from Siemens, and a project from NEC.

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

## A    EXPERIMENT DETAILS AND ADDITIONAL RESULTS

### A.1    SYNTHETIC DATA

**Comparison with Other POVI Methods**   We present posterior approximations obtained by weight-space and function-space versions of other POVI methods. The simulation setup is the same as in Section 5.1. As shown in Figure 4, function-space methods provide improvement in all cases, avoid the degenerate behavior of weight-space methods.

**Experiments with Increasing Model Complexity**   To obtain a better understanding of the degenerate behavior of weight-space POVI methods, we repeat the experiment with increasingly complex models. Specifically, we repeat the experiment while varying the number of hidden units in each layer from 5 to 100. Other settings are the same as Section 5.1. The posteriors are shown in Figure 5. We can see that weight space methods provide accurate posterior estimates when the number of weights is small, and degenerate gradually as model complexity increases, eventually all particles collapse into a single function. On the contrary, function space methods produce stable approximations all the time.

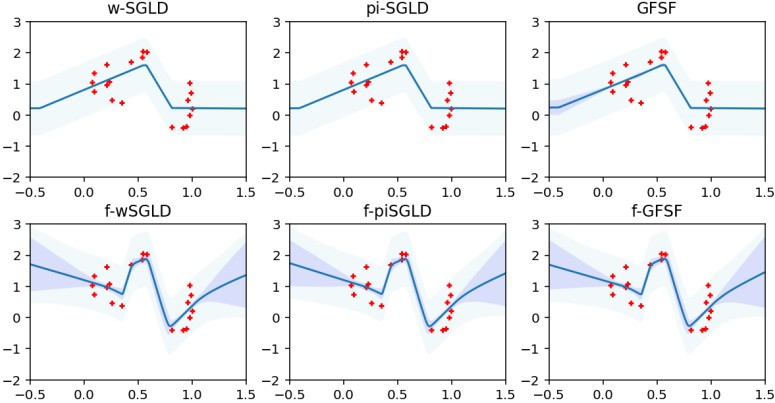

Figure 4: Posterior approximations obtained by weight space (up) and function space (down) variants of other POVI methods.

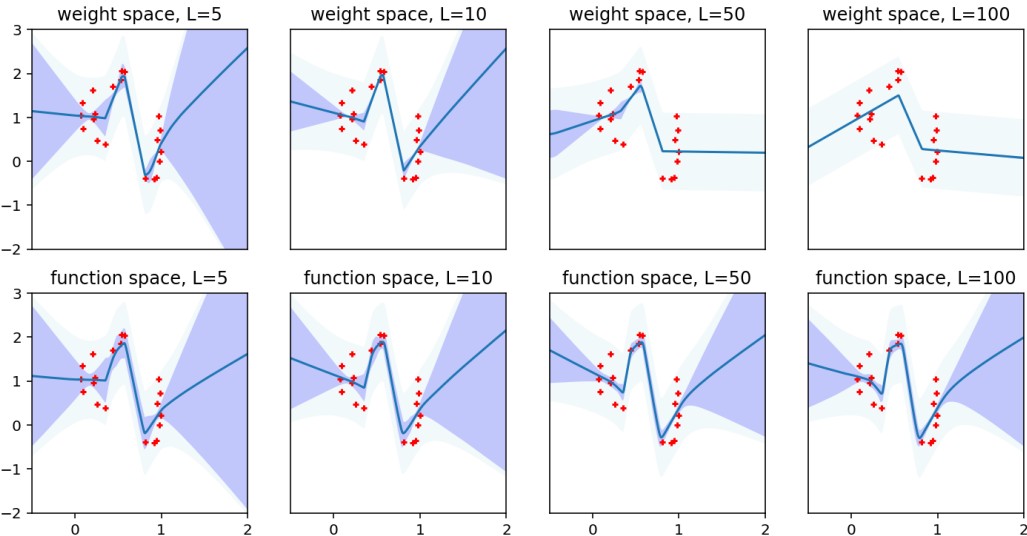

Figure 5: Posterior approximations on increasingly complex models. $L$ denotes the number of hidden units in each layer. We can see a clear trend of degeneration for weight-space method.

## A.2 UCI DATASETS

### A.2.1 EXPERIMENT DETAILS IN 5.2.1

For our method in all datasets, we use the AdaM optimizer with a learning rate of 0.004. For datasets with fewer than 1000 samples, we use a batch size of 100 and train for 500 epochs. For the larger datasets, we set the batch size to 1000, and train for 1000 epochs. We use a 90-10 random train-test split, repeated for 20 times, except for Protein in which we use 5 replicas. For our method and weight-space POVI methods, we use 20 particles. We use the RBF kernel, with the bandwidth chosen by the median trick (Liu & Wang, 2016).

For our method, we approximate the function-space prior with GP, and separate the mini-batch used for prior gradient and other parts in $\mathbf{v}$, as discussed in the main text. We construct the multivariate normal prior approximation with 40 draws and a batch size of 4.

### A.2.2 FULL RESULTS WITH DIFFERENT POVI PROCEDURES

In this section, we provide full results on UCI datasets, using weight-space and function-space versions of different POVI procedures. The experiment setup is the same as Section 5.2.1. The predictive RMSE and NLL are presented in Table 4 and 5, respectively. We can see that for all POVI methods, function space variants provide substantial improvement over their weight-space counterparts.

Table 4: Average test RMSE on UCI datasets. Bold indicates statistically significant best results ($p < 0.05$ with t-test).

| Dataset | Weight Space | | | Function Space (Ours) | | |
|---|---|---|---|---|---|---|
| | SVGD | w-SGLD | pi-SGLD | SVGD | w-SGLD | pi-SGLD |
| Boston | $2.96 \pm 0.10$ | $\mathbf{2.84 \pm 0.15}$ | $\mathbf{2.84 \pm 0.15}$ | $\mathbf{2.61 \pm 0.12}$ | $\mathbf{2.62 \pm 0.12}$ | $\mathbf{2.62 \pm 0.12}$ |
| Concrete | $5.32 \pm 0.10$ | $5.51 \pm 0.10$ | $5.49 \pm 0.10$ | $\mathbf{4.73 \pm 0.13}$ | $\mathbf{4.78 \pm 0.14}$ | $\mathbf{4.77 \pm 0.13}$ |
| Kin8nm | $0.09 \pm 0.00$ | $0.07 \pm 0.00$ | $0.07 \pm 0.00$ | $\mathbf{0.07 \pm 0.00}$ | $0.07 \pm 0.00$ | $0.07 \pm 0.00$ |
| Naval | $0.00 \pm 0.00$ | $\mathbf{0.00 \pm 0.00}$ | $\mathbf{0.00 \pm 0.00}$ | $0.00 \pm 0.00$ | $\mathbf{0.00 \pm 0.00}$ | $\mathbf{0.00 \pm 0.00}$ |
| Power | $3.94 \pm 0.03$ | $3.97 \pm 0.03$ | $3.97 \pm 0.03$ | $\mathbf{3.80 \pm 0.03}$ | $\mathbf{3.83 \pm 0.03}$ | $\mathbf{3.83 \pm 0.03}$ |
| Protein | $4.61 \pm 0.01$ | $4.40 \pm 0.02$ | $4.40 \pm 0.02$ | $\mathbf{3.96 \pm 0.03}$ | $\mathbf{4.02 \pm 0.03}$ | $\mathbf{4.02 \pm 0.03}$ |
| Winered | $\mathbf{0.61 \pm 0.01}$ | $0.63 \pm 0.01$ | $0.63 \pm 0.01$ | $\mathbf{0.61 \pm 0.01}$ | $\mathbf{0.61 \pm 0.01}$ | $\mathbf{0.61 \pm 0.01}$ |
| Yacht | $0.86 \pm 0.05$ | $0.95 \pm 0.07$ | $0.93 \pm 0.07$ | $\mathbf{0.63 \pm 0.06}$ | $\mathbf{0.63 \pm 0.06}$ | $\mathbf{0.63 \pm 0.06}$ |

Table 5: Average test NLL on UCI datasets. Bold indicates best results.

| Dataset | Weight Space | | | Function Space (Ours) | | |
|---|---|---|---|---|---|---|
| | SVGD | w-SGLD | pi-SGLD | SVGD | w-SGLD | pi-SGLD |
| Boston | $\mathbf{2.50 \pm 0.03}$ | $\mathbf{2.52 \pm 0.07}$ | $\mathbf{2.52 \pm 0.07}$ | $2.47 \pm 0.08$ | $2.52 \pm 0.09$ | $2.51 \pm 0.08$ |
| Concrete | $3.08 \pm 0.02$ | $3.15 \pm 0.03$ | $3.15 \pm 0.03$ | $\mathbf{2.96 \pm 0.04}$ | $\mathbf{3.02 \pm 0.04}$ | $\mathbf{3.01 \pm 0.04}$ |
| Kin8nm | $-0.98 \pm 0.01$ | $-1.20 \pm 0.01$ | $-1.20 \pm 0.01$ | $\mathbf{-1.24 \pm 0.00}$ | $\mathbf{-1.25 \pm 0.00}$ | $\mathbf{-1.25 \pm 0.00}$ |
| Naval | $-4.09 \pm 0.01$ | $-6.33 \pm 0.02$ | $-6.39 \pm 0.02$ | $-5.92 \pm 0.03$ | $\mathbf{-6.43 \pm 0.04}$ | $\mathbf{-6.45 \pm 0.05}$ |
| Power | $2.79 \pm 0.01$ | $2.80 \pm 0.01$ | $2.80 \pm 0.01$ | $\mathbf{2.76 \pm 0.01}$ | $\mathbf{2.76 \pm 0.01}$ | $\mathbf{2.76 \pm 0.01}$ |
| Protein | $2.95 \pm 0.00$ | $2.90 \pm 0.00$ | $2.90 \pm 0.00$ | $\mathbf{2.79 \pm 0.01}$ | $\mathbf{2.81 \pm 0.01}$ | $\mathbf{2.81 \pm 0.01}$ |
| Winered | $\mathbf{0.93 \pm 0.01}$ | $0.97 \pm 0.01$ | $0.97 \pm 0.01$ | $\mathbf{0.92 \pm 0.02}$ | $\mathbf{0.93 \pm 0.02}$ | $0.94 \pm 0.02$ |
| Yacht | $1.23 \pm 0.04$ | $1.39 \pm 0.07$ | $1.37 \pm 0.07$ | $\mathbf{1.01 \pm 0.06}$ | $\mathbf{0.99 \pm 0.06}$ | $\mathbf{1.00 \pm 0.06}$ |

### A.2.3 COMPARISON WITH OTHER METHODS

In this section, we provide comparison to a few other strong baselines in BNN inference.

**MNF and MCBN** We first present comparisons with multiplicative normalizing flow (MNF; Louizos & Welling, 2017) and Monte-Carlo batch normalization (MCBN; Azizpour et al., 2018). We make the experiment setups consistent with Azizpour et al. (2018), and cite the baseline results from their work. The model is a BNN with 2 hidden layer, each with 50 units for the smaller datasets, and 100 for the protein dataset. We use $10\%$ data for test, and hold out an additional fraction of $10\%$ data to determine the optimal number of epochs. The hyperparameters and optimization scheme for our model is the same as in Section 5.2.1. The results are presented in Table 6. We can see that in most cases, our algorithm performs better than the baselines.

**VIP**  We present comparison to the variational implicit process (VIP; Ma et al., 2018). We follow the experiment setup in their work, and cite results from it. The model is a BNN with 2 hidden layer, each with 10 units. For our method, we follow the setup in Section 5.2.1. We note that this is not entirely a fair comparison, as the method in Ma et al. (2018) also enables model hyper-parameter learning, while for our method we keep the hyper-parameters fixed. However, as shown in Table 7, our method still compares even or favorably to them, outperforming them in NLL in 5 out of 9 datasets. This supports our hypothesis in Section 4 that the linear combination posterior used in Ma et al. (2018) limits their flexibility.

Table 6: Average test RMSE and NLL following the setup of Azizpour et al. (2018). Bold indicates best results.

| Dataset | Test RMSE | | | Test NLL | | |
|---|---|---|---|---|---|---|
| | MCBN | MNF | f-SVGD | MCBN | MNF | f-SVGD |
| Boston | $\mathbf{2.75 \pm 0.05}$ | $2.98 \pm 0.06$ | $2.95 \pm 0.08$ | $\mathbf{2.38 \pm 0.02}$ | $2.51 \pm 0.06$ | $2.57 \pm 0.07$ |
| Concrete | $\mathbf{4.78 \pm 0.09}$ | $6.57 \pm 0.04$ | $\mathbf{5.03 \pm 0.10}$ | $3.45 \pm 0.11$ | $3.35 \pm 0.04$ | $\mathbf{3.16 \pm 0.05}$ |
| Kin8nm | $0.07 \pm 0.00$ | $0.09 \pm 0.00$ | $\mathbf{0.06 \pm 0.00}$ | $-1.21 \pm 0.01$ | $-1.04 \pm 0.00$ | $\mathbf{-1.33 \pm 0.00}$ |
| Power | $3.74 \pm 0.01$ | $4.19 \pm 0.01$ | $\mathbf{3.50 \pm 0.01}$ | $2.75 \pm 0.00$ | $2.86 \pm 0.01$ | $\mathbf{2.67 \pm 0.00}$ |
| Protein | $3.66 \pm 0.01$ | $4.10 \pm 0.01$ | $\mathbf{3.36 \pm 0.03}$ | $2.73 \pm 0.00$ | $2.83 \pm 0.01$ | $\mathbf{2.56 \pm 0.01}$ |
| Winered | $0.62 \pm 0.00$ | $\mathbf{0.61 \pm 0.00}$ | $0.63 \pm 0.01$ | $0.95 \pm 0.01$ | $\mathbf{0.93 \pm 0.00}$ | $0.98 \pm 0.01$ |
| Yacht | $1.23 \pm 0.05$ | $2.13 \pm 0.05$ | $\mathbf{0.85 \pm 0.06}$ | $1.39 \pm 0.03$ | $1.96 \pm 0.05$ | $\mathbf{1.03 \pm 0.03}$ |

Table 7: Average test RMSE and NLL on UCI datasets, following the setup in Ma et al. (2018). Bold indicates best results.

| | NLL | | RMSE | |
|---|---|---|---|---|
| | VIP-BNN | f-SVGD | VIP-BNN | f-SVGD |
| Boston | $2.45 \pm 0.04$ | $\mathbf{2.33 \pm 0.05}$ | $2.88 \pm 0.14$ | $\mathbf{2.58 \pm 0.12}$ |
| Concrete | $3.02 \pm 0.02$ | $\mathbf{2.93 \pm 0.02}$ | $4.81 \pm 0.13$ | $\mathbf{4.63 \pm 0.12}$ |
| Energy | $\mathbf{0.60 \pm 0.03}$ | $0.69 \pm 0.03$ | $0.45 \pm 0.01$ | $0.43 \pm 0.01$ |
| Kin8nm | $-1.12 \pm 0.01$ | $-1.11 \pm 0.01$ | $0.07 \pm 0.00$ | $0.07 \pm 0.00$ |
| Power | $2.92 \pm 0.00$ | $\mathbf{2.76 \pm 0.00}$ | $4.11 \pm 0.05$ | $\mathbf{3.78 \pm 0.02}$ |
| Protein | $2.87 \pm 0.00$ | $\mathbf{2.85 \pm 0.00}$ | $4.25 \pm 0.07$ | $\mathbf{3.95 \pm 0.02}$ |
| Winered | $0.97 \pm 0.02$ | $\mathbf{0.89 \pm 0.01}$ | $0.64 \pm 0.01$ | $\mathbf{0.61 \pm 0.01}$ |
| Yacht | $\mathbf{-0.02 \pm 0.07}$ | $1.17 \pm 0.01$ | $0.32 \pm 0.06$ | $0.67 \pm 0.07$ |
| Naval | $\mathbf{-5.62 \pm 0.04}$ | $-5.41 \pm 0.10$ | $0.00 \pm 0.00$ | $0.00 \pm 0.00$ |

### A.2.4  FURTHER EXPERIMENTS

As suggested by the reviewers, we add further experiments to evaluate our method.

**Convergence Properties of Our Method**  We demonstrate the stability of our procedure, by presenting the negative log-likelihood on held-out data during a typical run. The experiment setup follows (Azizpour et al., 2018), and the result is plotted in Figure 6. As shown in the figure, the training process of our procedure is fairly stable.

**Benchmark on a Narrow Architecture**  To balance our discussion on weight-space POVI methods, we present a benchmark on a narrower BNN model. More specifically, we use the experiment setup in (Ma et al., 2018), which uses 10 hidden units in each layer. Other settings are summarized in the previous subsection. We compare f-SVGD with SVGD, and include the results for mean-field VI and MC dropout with $\alpha$-divergence in (Ma et al., 2018) for reference. The results are presented in Table 8. We can see that weight-space POVI is a strong baseline under priors with simpler architectures. However, as the issue of over-parameterization still presents, the function-space method still outperform it by a significant margin.

Table 8: Comparison with other baselines under a narrower network prior on some UCI datasets. boldface indicates best results.

| | NLL | | | | RMSE | | | |
|---|---|---|---|---|---|---|---|---|
| | MFVI | alpha-dropout | SVGD | f-SVGD | MFVI | alpha-dropout | SVGD | f-SVGD |
| Boston | 2.76 (0.04) | 2.45 (0.02) | **2.42 (0.07)** | **2.33 (0.05)** | 3.85 (0.22) | 3.06 (0.09) | **2.77 (0.20)** | **2.58 (0.12)** |
| Kin8mn | -0.81 (0.01) | -0.92 (0.02) | **-1.11 (0.01)** | **-1.11 (0.00)** | 0.10 (0.00) | 0.10 (0.00) | **0.08 (0.00)** | **0.08 (0.00)** |
| Power | 2.83 (0.01) | 2.81 (0.00) | 2.79 (0.01) | **2.76 (0.00)** | 4.11 (0.04) | 4.08 (0.00) | 3.95 (0.02) | **3.78 (0.02)** |
| Protein | 3.00 (0.00) | 2.90 (0.00) | 2.87 (0.00) | **2.85 (0.00)** | 4.88 (0.04) | 4.46 (0.00) | 4.29 (0.02) | **3.95 (0.02)** |

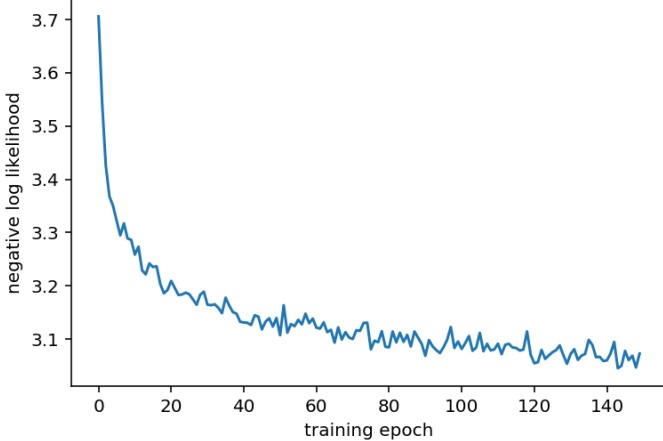

Figure 6: Heldout NLL as a function of training time (in epochs) during a sample run on the Concrete dataset.

### A.3 ADVERSARIAL EXAMPLES

**MNIST Experiment Details** We follow the setup in Li & Gal (2017). We use a feed-forward network with ReLU activation and 3 hidden layers, each with 1000 units. For both SVGD and f-SVGD, we use the AdaM optimizer with learning rate $5 \times 10^{-4}$; we use a batch size of 1000 and train for 1000 epochs. The attack method is the targeted iterated FGSM with $\ell^\infty$ norm constraint, i.e. for $t = 1, \ldots, T$, set

$$x_{adv}^{(t)} := x_{adv}^{(t-1)} + \epsilon \cdot \text{sgn}(\nabla \log \mathbb{E}_{q(f)}[p(y = 0 | x_{adv}^{(t-1)}, f)]),$$

where $\epsilon = 0.01$, and $x_{adv}^{(0)}$ denotes the clean input. We remove clean images labeled with "0".

For our method and weight-space SVGD, we use 20 particles, and scale the output logits to have a prior variance of 10. We use an additional 40 particles to generate 8-dimensional prior samples.

A few notes on applying our method to classification problems: it is important to use the *normalized logits* as $f$, so that the model is fully identified; also as for classification problems, $f(x)$ is already high-dimensional for a single data point $x$, one should down-sample $f(x)$ within each $x$. We choose to always include the true label in the down-sampled version, for labeled samples, and share negative labels across different $x$.

Test accuracy and log likelihood on clean samples are reported in Table 9.

Table 9: MNIST: Test accuracy and NLL on clean samples

|             | MC-dropout | SVGD  | f-SVGD |
|-------------|------------|-------|--------|
| accuracy    | 0.983      | 0.970 | 0.984  |
| average NLL | 0.075      | 0.109 | 0.065  |

**CIFAR Experiment Details** We use the ResNet-32 architecture, defined in He et al. (2016b), and uses the same training scheme. We use 8 particles for our method and the ensemble method. For our method, we use 32 particles to generate 6-dimensional prior samples.

The ResNet architecture consists of batch normalization layers, which needs to be computed with the full batch of input. We approximate it with a down-sampled batch, so that more prior samples can be produced more efficiently. In our implementation, we down-sample the input batch to $1/4$ of its original size.

The test accuracy and log likelihood on clean samples are reported in Table 10. Our method out-performs the single point estimate, but is slightly worse than the ensemble prediction. Performance

drop on clean samples is common that models obtained by adversarially robust methods (cf. e.g. Liao et al., 2017).

Table 10: CIFAR-10: Test accuracy and NLL on clean samples

|  | single | ensemble | f-SVGD |
|---|---|---|---|
| accuracy | 0.925 | 0.937 | 0.934 |
| average NLL | 0.376 | 0.203 | 0.218 |

## A.4 CONTEXTUAL BANDIT

Contextual bandit is a classical online learning problem. The problem setup is as follows: for each time $t = 1, 2, \cdots, N$, a context $s_t \in \mathcal{S}$ is provided to the online learner, where $\mathcal{S}$ denotes the given context set. The online learner need to choose one of the $K$ available actions $I_t \in \{1, 2, \cdots, K\}$ based on context $s_t$, and get a (stochastic) reward $\ell_{I_t, t}$. The goal of the online learner is to minimize the pseudo-regret

$$\overline{R}_n^{\mathcal{S}} = \max_{g: \mathcal{S} \to \{1, 2, \cdots, K\}} \mathbb{E}\left[\sum_{t=1}^n \ell_{g(s_t), t} - \sum_{t=1}^n \ell_{I_t, t}\right]. \tag{5}$$

where $g$ denotes the mapping from context set $\mathcal{S}$ to available actions $\{1, 2, \cdots, K\}$. Pseudo-regret measures the regret of not following the best $g$, thus pseudo-regret is non-negative, and minimize the pseudo-regret is equal to find such the best $g$.

For contextual bandits with non-adversarial rewards, Thompson sampling (a.k.a. posterior sampling; Thompson, 1933) is a classical algorithm that achieves state-of-the-art performance in practice (Chapelle & Li, 2011). Denote the underlying ground-truth reward distribution of context $s$ and action $I_t$ as $\nu_{s, I_t}$. In Thompson sampling, we place a prior $\mu_{s, i, 0}$ on reward for context $s$ and action $i$, and maintain $\mu_{s, i, t}$, the corresponding posterior distribution at time $t$. For each time $t = 1, 2, \ldots, N$, Thompson sampling selects action by

$$I_t \in \underset{i = \{1, 2, \cdots, K\}}{\arg \max} \hat{\ell}_{i, t}, \quad \hat{\ell}_{i, t} \sim \mu_{s, i, t}.$$

The corresponding posterior is then updated with the observed reward $\ell_{I_t, t}$. The whole procedure of Thompson sampling is shown in Algorithm 2.

---
**Algorithm 2** Thompson Sampling
---
    **Input:** Prior distribution $\mu_{s, i, 0}$, time horizon $N$
    **for** time $t = 1, 2, \cdots, N$ **do**
        Observe context $s_t \in \mathcal{S}$
        Sample $\hat{\ell}_{i, t} \sim \mu_{s, i, t}$.
        Select $I_t \in \arg\max_{i = \{1, 2, \cdots, K\}} \hat{\ell}_{i, t}$ and get $\ell_{I_t, t}$
        Update the posterior of $\mu_{s_t, I_t, t+1}$ with $\ell_{I_t, t}$.
    **end for**
---

Notice that contextual bandit problems always face the *exploration-exploitation dilemma*. Exploration should be appropriate, otherwise, we can either exploit too much sub-optimal actions or explore too much meaningless actions. Thompson sampling addresses this issue by each time selecting the actions greedy with the sampled rewards, which is equal to selecting action $i$ with the probability that $i$ can get the highest reward under context $s_t$. This procedure need an accurate posterior uncertainty. Either over-estimate or under-estimate the posterior uncertainty can lead to the failure of balancing exploration and exploitation, which further lead to the failure of Thompson sampling.

Here, we focus on two benchmark contextual bandit problems, called *mushroom* and *wheel*.

**Mushroom Bandit** In mushroom, we use the data from mushroom dataset (Schlimmer, 1981), which contains 22 attributes per mushroom and two classes: poisonous and safe. Eating a safe mushroom provides reward +5. Eating a poisonous mushroom delivers reward +5 with probability 1/2 and reward -35 otherwise. If the agent does not eat a mushroom, then the reward is 0. We run 50000 rounds in this problem.

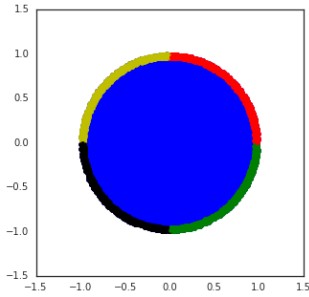

Figure 7: Visualization of the wheel bandit, taken from (Riquelme et al., 2018). Best viewed in color.

**Wheel Bandit** Wheel bandit is a synthetic problem that highlights the need for exploration. Let $\delta \in (0, 1)$ be an "exploration parameter". Context $X$ is sampled uniformly at random in the unit circle in $\mathbb{R}^2$. There are $k = 5$ possible actions: the first action results in a constant reward $\ell_1 \sim \mathcal{N}(\mu_1, \sigma^2)$; the reward corresponding to other actions is determined by $X$:

- For contexts inside the blue circle in Figure 7, i.e. for $X$ s.t. $\|X\| \leq \delta$, the other four actions all result in a suboptimal reward $\mathcal{N}(\mu_2, \sigma^2)$ for $\mu_2 < \mu_1$.
- Otherwise, one of the four contexts becomes optimal depend on the quarter $X$ is in. The optimal action results in a reward of $\mathcal{N}(\mu_3, \sigma^2)$ for $\mu_3 \gg \mu_1$, and other actions still have the reward $\mathcal{N}(\mu_2, \sigma^2)$.

As the probability that $X$ corresponds to a high reward is $1 - \delta^2$, the need for exploration increases as $\delta$ increases, and it is expected that algorithm with poorly calibrated uncertainty will stuck in choosing the suboptimal action $a_1$ in these regions. Such a hypothesis is confirmed in (Riquelme et al., 2018), making this bandit a particularly challenging problem. In our experiments, we use 50000 contexts, and set $\delta = 0.95$.

**Experiment Setup** The model for neural networks is a feed-forward network with 2 hidden layers, each with 100 units. The hyper-parameters for all models are tuned on the mushroom bandit, and kept the same for the wheel bandit. For the baselines, we use the hyper-parameters provided in (Riquelme et al., 2018). The main difference from (Riquelme et al., 2018) is that we use 20 replicas for bootstrap. For our method, we also use 20 particles.

## B    ON ALTERNATIVE KERNELS FOR WEIGHT-SPACE POVI

Many POVI methods use a kernel to make the gradient flow well-defined for discrete distributions. A natural question is whether we could design the kernel carefully to alleviate the problem of over-parameterization and high dimensionality. This idea is tempting: intuitively, for most POVI methods listed in Table 1, the kernel defines a *repulsive force term*, which push a particle away from its "most similar" neighbors. A better choice of the kernel makes the similarity measure more sensible. In this section, we will list candidates for such a kernel, and show that they do not work in general.

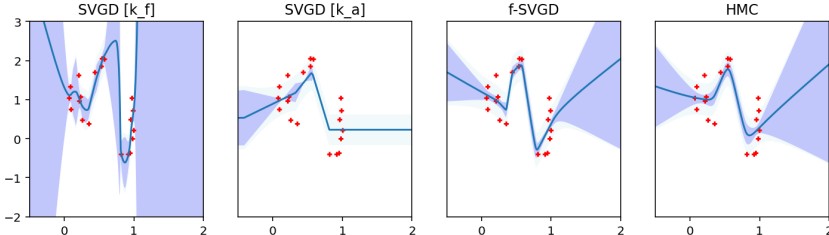

Figure 8: Posterior approximations with weight-space SVGD and alternative kernels. `k_f` corresponds to the function-value kernel, and `k_a` the activation kernel. We include the results for f-SVGD and HMC for reference.

Table 11: Test NLL on UCI datasets for SVGD with alternative kernels.

|  | Weight-Space SVGD | | | Function-Space SVGD |
|---|---|---|---|---|
|  | RBF | $k_a$ | $k_f$ | |
| Boston | $2.50 \pm 0.03$ | $2.50 \pm 0.07$ | $2.49 \pm 0.06$ | $2.47 \pm 0.08$ |
| Yacht | $1.23 \pm 0.04$ | $1.35 \pm 0.06$ | $1.20 \pm 0.07$ | $\mathbf{1.01 \pm 0.06}$ |
| Concrete | $3.08 \pm 0.02$ | $3.12 \pm 0.03$ | $3.11 \pm 0.03$ | $\mathbf{2.96 \pm 0.04}$ |

**Evaluation Setup**    For kernels considered in this section, we evaluate their empirical performance on the synthetic dataset and the UCI regression datasets, following the same setup as in the main text. We test on two POVI methods: SVGD and w-SGLD. As results are similar, we only report the results for SVGD for brevity. The results are presented in Figure 8 and Table 11.

**The "Function Value" Kernel**    Similar to our proposed method, one could define a *weight-space* kernel on function values, so it directly measures the difference of regression functions; i.e.

$$k_f(\theta^{(1)}, \theta^{(2)}) := \mathbb{E}_{\mathbf{x} \sim \mu}[k(f(\mathbf{x}; \theta^{(1)}), f(\mathbf{x}; \theta^{(2)}))], \qquad (6)$$

where $k$ is an ordinary kernel on $\mathbb{R}^B$ (e.g. the RBF kernel), $\mathbf{x} \in \mathcal{X}^B$, and $\mu$ is an arbitrary measure supported on $\mathcal{X}^B$. We first remark that $k_f$ is not positive definite (p.d.) due to over-parameterization, and there is no guarantee that weight-space inference with $k_f$ will converge to the true posterior in the asymptotic limit[8]. Furthermore, it does not improve over the RBF kernel empirically: predictive performance does not improve, and it drastically overestimates the epistemic uncertainty on the synthetic data.

To understand the failure of this kernel, take SVGD as an example: the update rule $\theta_{\ell+1}^{(i)} \leftarrow \theta_\ell^{(i)} - \epsilon_\ell \mathbf{v}(\theta_\ell^{(i)})$ is defined with

$$-\mathbf{v}(\theta^{(i)}) = \frac{1}{n} \sum_{j=1}^{n} \left( \underbrace{\mathbf{K}_{ij} \nabla_{\theta^{(j)}} \log p(\theta^{(j)}|\mathbf{x})}_{\mathrm{grad}_{ij}} + \underbrace{\nabla_{\theta^{(j)}} \mathbf{K}_{ij}}_{\mathrm{rf}_{ij}} \right).$$

Finite-sample SVGD is usually understood as follows (Liu & Wang, 2016): the first term follows a smoothed gradient direction, which pushes the particles towards high-probability region in posterior;

---

[8]E.g. Liu & Wang (2016) requires a p.d. kernel for such guarantees.

the second term acts as a repulsive force, which prevents particles from collapse together. However, for inference with non positive definite kernels such as $k_f$, it is less clear if these terms still play the same role:

1. When there are particles corresponding to the same regression function, their gradient for log posterior will be averaged in grad. This is clearly undesirable, as these particles can be distant to each other in the weight space, so their gradient contains little learning signal for each other.

2. While for stationary kernels, the repulsive force

$$\mathrm{rf}_{ij} = \frac{1}{n}\sum_j \nabla_{\theta^{(j)}}\mathbf{K}_{ij} = -\frac{1}{n}\sum_j \nabla_{\theta^{(i)}}\mathbf{K}_{ij}$$

drives particle $i$ away from other particles, this equality does not hold for $k_f$ which is non-stationary. Furthermore, in such summation over $\nabla_{\theta^{(j)}}\mathbf{K}_{ij}$, as $\nabla_{\theta^{(j)}}\mathbf{K}_{ij} = \nabla_{f(\mathbf{x};\theta^{(j)})}\mathbf{K}_{ij}\nabla_{\theta^{(j)}}f(\mathbf{x};\theta^{(j)})$, assuming $\nabla_{\theta^{(j)}}f(\mathbf{x};\theta^{(j)})$ is of the same scale for $j$, $\nabla_{\theta^{(j)}}f(\mathbf{x};\theta^{(j)})$ will contribute to particle $i$'s repulsive force most if the function-space repulsive force, $\nabla_{f(\mathbf{x};\theta^{(j)})}\mathbf{K}_{ij}$ is the largest. However, unlike in identifiable models, there is no guarantee that this condition imply $\theta^{(j)}$ is sufficiently close to $\theta^{(i)}$, and mixing their gradient could be detrimental for the learning process.

As similar terms also exist in other POVI methods, such an argument is independent to the choice of POVI methods. Also note that both parts of this argument depends on the over-parameterization property of the model, and the degeneracy (not being p.d.) of the kernel.

**The Activation Kernel**  Another kernel that seems appropriate for BNN inference can be defined using network layer activations. Specifically, let $h(\mathbf{x};\theta)$ be the activations of all layers in the NN parameterized by $\theta$, when fed with a batch of inputs $\mathbf{x}$. The kernel is defined as

$$k_a(\theta^{(1)}, \theta^{(2)}) := \mathbb{E}_{\mathbf{x}\sim\mu}[k(h(\mathbf{x};\theta^{(1)}), h(\mathbf{x};\theta^{(2)})))].$$

This kernel is positive definite if the batch size $B$ is sufficiently large, but it does not work either: predictive performance is worse than weight-space SVGD with RBF kernel, and as shown in Figure 8, it also suffers from the collapsed prediction issue. Intuitively, our argument on over-parameterization in Section 2 should apply to all positive definite kernels.

In conclusion, constructing sensible kernels for BNN in weight-space POVI is non-trivial; on the other hand, our proposed algorithm provides a more natural solution, and works better in practice.

## C    IMPACT OF PARAMETRIC AND STOCHASTIC APPROXIMATIONS

Our algorithm consists of several approximations. Assuming the function-space prior gradient is known, there remains two approximations, namely the parametric approximation to function particles (Section 3.1.1), and the stochastic approximation to particle update (Section 3.1.2). In this section, we examine the impact of them empirically, by comparing them to an exact implementation of the function-space POVI algorithm in a toy setting.

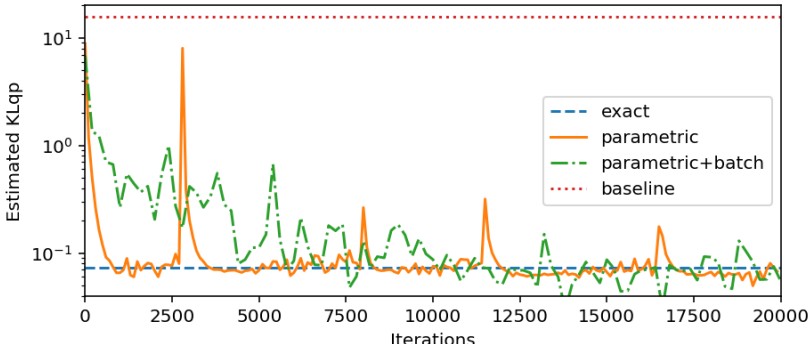

Figure 9: Estimated KL $(q\|p)$ w.r.t. training iterations, for f-SVGD using different sets of approximations. A baseline value is presented to help understanding the scale of KL divergence in this experiment; see below for details.

**Experiment Setup**    To make exact simulation of the function-space POVI procedure possible, we consider a 1-D regression problem on a finite-dimensional function space, with a closed form GP prior. Specifically, we set $\mathcal{X} := \mathbf{X}_{\text{train}} \cup \mathbf{X}_{\text{test}} := \{-2, -1.8, -1.6, \ldots, +2\} \cup \{1.7, 1.9, 2.1\}$. The training targets are generated by $y_i \sim \mathcal{N}(\sin(x_i), 0.1^2)$. The function-space prior is a GP prior with zero mean and a RBF kernel, which has a bandwidth of $0.5$. The true posterior can be computed in closed form (Rasmussen, 2004):

$$\mathbf{f}(\mathbf{X}_{\text{test}})|\mathbf{X}_{\text{train}}, \mathbf{Y}_{\text{train}} \sim \mathcal{N}(\mu, \Sigma), \qquad\qquad \text{where}$$
$$\mu := \mathbf{K}_{tr}(\mathbf{K}_{rr} + \sigma^2 I)^{-1}(\mathbf{Y}_{\text{train}}),$$
$$\Sigma := \mathbf{K}_{tt} - \mathbf{K}_{tr}(\mathbf{K}_{rr} + \sigma^2 I)^{-1}\mathbf{K}_{tr}^{\top},$$

and $\mathbf{K}_{tt}, \mathbf{K}_{tr}, \mathbf{K}_{rr}$ denote the gram matrices $k(\mathbf{X}_{\text{test}}, \mathbf{X}_{\text{test}}), k(\mathbf{X}_{\text{test}}, \mathbf{X}_{\text{train}})$ and $k(\mathbf{X}_{\text{train}}, \mathbf{X}_{\text{train}})$, respectively. We consider three versions of f-SVGD implementation:

- An "exact" implementation, which treats $f(\mathcal{X})$ as the parameter space and directly applies SVGD. It is tractable in this experiment, as $f(\mathcal{X})$ is finite dimensional. The only approximation errors are due to discretization and particle approximation. It is known that as step-size approach 0, and number of particles approaches infinity, this method recovers the true posterior exactly (Liu et al., 2017).
- f-SVGD with the parametric approximation, i.e. the algorithm described in Section 3.1.1.
- f-SVGD with the parametric approximation, and the mini-batch approximation, i.e. the algorithm described in Section 3.1.2.

To understand the scale of the KL divergence used here, we also report its value using a baseline posterior approximation $q$, defined as the GP posterior conditioned on a *down-sampled training set*, $\tilde{\mathbf{X}}_{\text{train}} := \{-2, -1.6, \ldots, +2\}$.

We use $1,000$ particles and a step size of $10^{-3}$. We run each version of f-SVGD for 20,000 iterations, and report the exclusive KL divergence of test predictions, KL $(q[f(\mathbf{X}_{\text{test}})]\|p[f(\mathbf{X}_{\text{test}})])$, where $q(f(\cdot))$ is approximated with a multivariate normal distribution.

The result is presented in Figure 9. We can see that even with 1,000 particles, our introduced approximations has a negligible impact on the solution quality, compared to the (well-understood)

discretization and particle approximation in the original SVGD algorithm. Furthermore, the final algorithm in Section 3.1.2 has no convergence issues.

# D  THE RELATION BETWEEN FUNCTION-SPACE AND WEIGHT-SPACE POVI, AND FREQUENTIST ENSEMBLE TRAINING

Function-space POVI, weight-space POVI, and ensembled gradient descent share a similar form: they all maintain $K$ particles in the parameter space; in each iteration, they all compute a update vector for each particle according to some rule, and add it to the parameter. In this section, we discuss this connection in more detail, by comparing the update rule for each algorithm.

Consider a real-valued regression problem, where the likelihood model is $p(y|f(x)) := \mathcal{N}(y|f(x), \sigma_y^2)$. We choose SVGD as the base POVI algorithm. The $B$-*dimensional* kernel for f-SVGD is the RBF kernel with bandwidth $\sigma_k^2$. For all algorithms, denote the particles maintained at iteration $\ell$ as $\theta_\ell^1, \dots, \theta_\ell^K$. The three algorithms can all be written as

$$\theta_{\ell+1}^i := \theta_\ell^i + \epsilon \mathbf{u}_\ell^i,$$

for different choices of $\mathbf{u}$. We drop the subscript $\ell$ below, as the context is clear. Denote $\mathcal{J}^i := \left( \frac{\partial f_\ell^i(\mathbf{x}_b)}{\partial \theta_\ell^i} \right)^\top$, where $\mathbf{x}_b, \mathbf{y}_b$ is the mini-batch from training set drawn in step $\ell$; denote $f^i(\cdot) := f(\cdot; \theta^i)$.

**The Ensemble Method**  The *ensemble algorithm* computes $K$ maximum-a-posteriori estimates independently. The update rule is thus

$$\mathbf{u}_{\text{ens}}^i := \nabla_{\theta^i} \log p(\theta^i | \mathbf{X}, \mathbf{Y}) \tag{7}$$

$$\approx \frac{N}{B} \nabla_{\theta^i} \log p(\mathbf{y}_b | \theta^i, \mathbf{x}_b) + \nabla_{\theta^i} \log p(\theta^i) \tag{8}$$

$$= \underbrace{\mathcal{J}^i \left[ \frac{N}{B\sigma_y^2} (\mathbf{y}_b - f^i(\mathbf{x}_b)) \right]}_{\text{BP-ed error signal}} + \nabla_{\theta^i} \log p(\theta^i), \tag{9}$$

where $B$ is the batch size.

**f-SVGD**  The update rule of f-SVGD can be derived by plugging in $\mathbf{v}$ from Table 1 into (4). With a RBF kernel and the Gaussian likelihood model, its form is

$$\mathbf{u}_{\text{f}-\text{SVGD}}^i = -\mathcal{J}^i \mathbf{v}_{\text{SVGD}}(f_\ell^i) \tag{10}$$

$$= \sum_j \mathcal{J}^i \left[ \mathbf{K}_{ij} \nabla_{f^j(\mathbf{x})} \log p(f^j(\mathbf{x})|\mathbf{X}, \mathbf{Y}) + \nabla_{f^j(\mathbf{x})} \mathbf{K}_{ij} \right] \tag{11}$$

$$\approx \sum_j \left[ \mathbf{K}_{ij} \mathcal{J}^i \left( \frac{N}{B\sigma_y^2} (\mathbf{y}_b - f^j(\mathbf{x}_b)) + \nabla_{f^j(\mathbf{x})} \log p(f^j(\mathbf{x})) \right) + \underbrace{\frac{\mathbf{K}_{ij}}{\sigma_k^2} \mathcal{J}^i (f^i(\mathbf{x}) - f^j(\mathbf{x}))}_{\mathbf{fRF}_{ij}} \right]. \tag{12}$$

Denote $S_i := \sum_j \mathbf{K}_{ij}$, $\tilde{K}_{ij} := \mathbf{K}_{ij}/S_i$, we turn (12) into

$$S_i \mathcal{J}^i \sum_j \left[ \tilde{K}_{ij} \left( \frac{N}{B\sigma_y^2} (\mathbf{y}_b - f^j(\mathbf{x}_b)) + \nabla_{f^j(\mathbf{x})} \log p(f^j(\mathbf{x})) \right) + \frac{1}{S_i} \mathbf{fRF}_{ij} \right] \tag{13}$$

$$\propto \mathcal{J}^i \left\{ \frac{N}{B\sigma_y^2} \left( \mathbf{y}_b - \sum_j \tilde{K}_{ij} f^j(\mathbf{x}_b) \right) + \sum_j \tilde{K}_{ij} \nabla_{f^j(\mathbf{x})} \log p(f^j(\mathbf{x})) + \frac{1}{S_i} \mathbf{fRF}_{ij} \right\} \tag{14}$$

The similarity between (12) and (9) is now clear: the first term in (14) corresponds to the back-propagated error signal in (9); the second term in (14), the function-space prior gradient, plays the same role as the parametric prior gradient in (9); A newly added term, $\mathbf{fRF}_{ij}$ directly pushes $f^i(\mathbf{x})$ away from $f^j(\mathbf{x})$, ensuring the particles eventually constitutes a posterior approximation, instead of

collapsing into a single MAP estimate. In (14), all three terms act as function-space error signals, and get back-propagated to the weight space through $\mathcal{J}^i$.

Consequently, our algorithm has many desirable properties:

1. During the initial phase of training, as randomly initialized function particles are distant from each other, the repulsive force is relatively small, and particles will reach high-density regions rapidly like SGD;

2. the repulsive force term takes into account the prediction of different particles, and prevents them from collapsing into a single, potentially overfitted, mode; as its scale is determined by the principled POVI algorithm, the final particles constitute an approximation to the function-space posterior;

3. for large models, our algorithm could be easily parallelized following the model parallelism paradigm, as in each iteration, only function-space error signals need to be exchanged among particles, and the communication overhead is proportional to the batch size and number of particles.[9]

**SVGD with the function-value kernel**   Now we consider weight-space SVGD with the function value kernel (6), where both the finite-dimensional base kernel, and $\mu$, are chosen to be the same as in f-SVGD. Therefore, fix a set of samples $\mathbf{x}$, the evaluation on $(\theta^i, \theta^j)$ of (the stochastic approximation to) (6) equals $\mathbf{K}_{ij}$. Still, we denote it as $K_{ij}^{fv}$, as this kernel is a function of *weights*. The update rule is

$$\mathbf{u}_{\text{SVGD,fv}}^i := \sum_j \left[ \mathbf{K}_{ij}^{fv} \nabla_{\theta^j} \log p(\theta^j | \mathbf{X}, \mathbf{Y}) + \nabla_{\theta^j} \mathbf{K}_{ij}^{fv} \right] \tag{15}$$

$$= \sum_j \left[ \mathbf{K}_{ij}^{fv} \left( \frac{\partial \theta^j}{\partial f^j} \right) \nabla_{f^j} \log p(f^j | \mathbf{X}, \mathbf{Y}) + \left( \frac{\partial \theta^j}{\partial f^j} \right) \nabla_{f^j} \mathbf{K}_{ij}^{fv} \right] \tag{16}$$

$$\approx \sum_j \left\{ \mathbf{K}_{ij}^{fv} \mathcal{J}^j \left( \frac{N}{B\sigma_y^2} (\mathbf{y}_b - f^j(\mathbf{x}_b)) \right) + \nabla_{\theta^j} \log p(\theta^j) + \frac{\mathbf{K}_{ij}^{fv}}{\sigma_k^2} \mathcal{J}^j (f^i(\mathbf{x}) - f^j(\mathbf{x})) \right\}. \tag{17}$$

While (17) and (12) are very similar, the behavior of the resulted algorithms are drastically different. The key reason is that in (17), the Jacobian is for particle $j$, and the first and third term in (17) are gradients for particle $j$. But they are applied to particle $i$. As we have discussed in Appendix B, this issue of *gradient mixing* could be highly detrimental for inference on over-parameterized models.

**SVGD with p.d. kernel**   Lastly, we present the update rule for SVGD using general kernels for completeness. It is

$$\mathbf{u}_{\text{SVGD,p.d.}}^i \approx \sum_j \left\{ \mathbf{K}_{ij}^{fv} \mathcal{J}^j \left( \frac{N}{B\sigma_y^2} (\mathbf{y}_b - f^j(\mathbf{x}_b)) \right) + \nabla_{\theta^j} \log p(\theta^j) + \nabla_{\theta^j} \mathbf{K}_{ij}^{fv} \right\}. \tag{18}$$

Although it also mixes gradients from different particles, such a behavior may be less detrimental than in the function-value kernel case, as the mixing coefficient $\mathbf{K}_{ij}$ is usually based on similarity between network weights[10]. The issue of weight-space POVI using p.d. kernel is due to over-parameterization, as we have discussed in Section 2.

---

[9] for smaller models like those used in real-valued regression, many (i.e. hundreds of) particles could fit in a single GPU device, and the communication overhead is negligible; furthermore, as a single particle requires far less computational resource than the GPU has, the initial scaling curve could be sub-linear, i.e. using 20 particles will not be 20 times slower than fitting a single particle, as verified in Liu & Wang (2016).

[10] In practice, however, we may re-scale the kernel values to introduce sensible repulsive force (e.g. the "median trick" in RBF kernel serves this purpose, as mentioned in Liu & Wang (2016)). In this case, the gradient mixing effect will also be noticeable. We hypothesize that it could be part of the reason of the pathologies in weight-space methods; as a supporting evidence, we experimented SVGD on the random feature approximation to deep GP (Cutajar et al., 2016) on the synthetic dataset from Louizos & Welling (2016). Using a 3-layer deep GP with Arc kernel and 5 GPs per layer, all particles collapse to a *constant function*, which is *not the posterior mean* given by HMC. A quantitative characterization of the impact of gradient-mixing is beyond the scope of this work.

