# OpenReview forum: "Function Space Particle Optimization for Bayesian Neural Networks"
_ICLR.cc/2019/Conference_

### Official Review · AnonReviewer3 · 2018-11-01
**This is an interesting paper that seems to make an important contribution but its technical presentation needs to be improved. As of now, it is somewhat hard to appreciate how significant the proposed solution is. I hope the authors could help me clarify the below points so I can converge to a final rating.**

**Rating:** 7
**Confidence:** 3

**Review:**

PAPER SUMMARY:

This paper proposes a new POVI method for posterior inference in BNN. Unlike existing POVI techniques that optimize particles in the weight space which often yields sub-optimal results on BNN due to its over-parameterized nature, the new POVI method aims to maintain and update particles  directly on the space of regression functions to overcome this sub-optimal issue.

NOVELTY & SIGNIFICANCE:

In general, I am inclined to think that this paper has made an important contribution with very promising results but I still have doubts in the proposed solution technique (as detailed below) and am not able to converge to a final rating at this point.

TECHNICAL SOUNDNESS:

The authors claim that the new POVI technique operates directly on the function-space posterior to sidestep the over-parameterized issue of BNN but ultimately each function particle is still identified by a weight particle (as detailed in Eq. (2)). In terms of high-level ideas, I am not sure I understand the implied fundamental differences between this work and SVGD and how significant is it.

On the technical level, the key difference between the proposed work and SVGD seems to be the particle update equation in (2): The gradient flow is multiplied with the derivative of the BNN evaluated at the corresponding weight particle (in SVGD, the gradient flow was used alone). The authors then mentioned that this update rule results from minimizing the difference between f(X, theta) and f(X, theta) + \epsilon * v(f(., theta))(X). I do not follow this step -- please elaborate.

The theoretical justification that follows Eq. (3) is somewhat incoherent: What is \Epsilon(q(f(x)))? This has not been defined before or anywhere in the main text. Furthermore, the paragraph that follows the theoretical justification implies the computation of the gradient flow in (3) involves the likelihood term -- why is that?

In Algorithm 1, why do we sample from both the training set and some measure \mu? I am sure there must be a reason for this but I could not find it anywhere except for a short statement that "for convenience, we choose \mu in such a way that samples from \mu always consists a mini-batch from X". Please elaborate.

Will the proposed POVI converge?

CLARITY:

I think this paper has clarity issue with the technical exposition. The explanation tends to be very limited and even appear coherent at important points. For example, see
my 3rd point above.

---

> ### Author Response · Authors · 2018-11-22
> **Thanks for your feedback; summary of high-level idea and response to individual questions below (1/2)**
>
> Thank you for your positive and constructive feedback. We have incorporated them in the revision. As you expressed some confusion about our proposed method and contribution, we first present an informal but more high-level explanation; responses to your individual questions are presented following that.
>
> ####### our high-level idea, its difference to weight-space SVGD, and contribution #######
>
> As stated in Section 2 of the revision, the vanilla SVGD performs Bayesian inference of BNNs in the space of network weights (namely “weight-space inference”). While such a view is natural, our key observation is that it can make the problem unnecessarily hard. Namely, SVGD works by placing particles in high-probability regions, and at the same time making sure they are distant from each other, so they constitute a representative sample of the posterior. But in the weight-space posterior of BNN, there exists (at least) exponentially many numbers of posterior maximas that can be distant from each other, but corresponding to the same function. So a possible convergence point is where each SVGD particle occupies one of them; and in prediction, this posterior approximation does not improve over a single point estimate. In other words, a good approximation of the weight-space posterior isn’t necessarily good in function space.
>
> We propose to do SVGD in function space. The key difference is that distance metrics are now defined on functions, so networks corresponding to the same function are far less likely to be represented by multiple particles (see Remark 3.2 and Appendix D in our revision). This is non-trivial as the function space is infinite-dimensional, but we have built an effective approximation that is very easy to implement.
>
> We believe our contribution is significant, as
> (1) our proposal is very easy to implement, and performs extremely well in practice;
> (2) inference in over-parameterized models has attracted attention for long (see the first paragraph of Section 4). By shifting the focus to the prediction function instead of its specific parameterization, our work presents a novel solution to this problem.

---

> > ### Author Response · Authors · 2018-11-22
> > **Response to individual questions (2/2)**
> >
> > ####### Response to Technical Questions #######
> >
> > Q1. Elaboration of the update rule (2) (now Eq. (3) in revision):
> > A1: The update rule (3) corresponds to doing one step of gradient descent to minimize the squared distance between f(X;theta) and f(X;theta)+\epsilon*v(f(X;\theta)). We clarified this in Remark 3.1 in our revision.
> >
> > Q2. What is \Epsilon(q(f(x)))?
> > A2: Thanks for pointing out. We revised the text to clarify it. \mathcal{E}(q[f(x)]) is an energy functional defined on the space of (variational) distributions, which achieves its unique minimum at the true posterior p. Its exact form varies across different POVI methods. Please refer to [Chen et al., 2018] for the exact form.
> >
> > Q3. Why (3) (now Eq (4) in revision) involves the likelihood term:
> > A3: In fact, any kinds of gradient flow (GF) that corresponds to a Bayesian inference procedure will include model posterior, and subsequently through the Bayes' rule, the likelihood term. The GFs considered in this work are defined in the newly added Eq. (1), in which v is defined in Table 1. You can see there that they all include the posterior term. We have revised the paragraph you mentioned to make this clearer.
> >
> > Q4. why we sampled from both the training set and some measure \mu:
> > A4: Thanks for pointing out. In fact, this is a typo. The correct statement is that we only need to sample from \mu, which includes samples from the training set. We sincerely apologize for the confusion, and have fixed it in the revision. We have also revised Section 3.1.2 to clarify why we need samples from both the training set and another distribution with a full support. An intuitive explanation is:
> > (1) firstly, to make sure prediction interpolates and extrapolates, at least some components of \mu (recall it is defined on X^B, and a sample from \mu contains B points in input space) must visit the entire input space, i.e. have a full support. So these components can’t only include the training points, and we incorporated a full-support component for this purpose.
> > (2) if samples from \mu does not contain training points with a non-zero probability, we will have trouble in computing the likelihood, as it is only defined (in closed form) on training samples. Therefore, we added the training sample component.
> >
> > Q5. On whether the proposed method will converge:
> > A5: We have added convergence plots in Appendix A.2.4 and Appendix C. We can see that in practice, the proposed procedure is fairly robust, and has no convergence issues. Moreover, our algorithm is an approximation to an idealized algorithm: the simulation of the ``averaged gradient flow’’ in paragraph “Theoretical Justification” of Section 3.1.2. As we discussed there, simulation of the averaged GF converges to a unique optima (i.e., the true posterior).
> > Finally, like much previous work, we do introduce some approximations (e.g., stochastic approximations to the averaged GF and parametric approximation to particle functions). Developing a theory that simultaneously considers all approximations is very hard, so we present thorough empirical evidence: in Section 5.1, the approximate posterior obtained by our algorithm is of high quality; and in the rest of Section 5, the approximation works well in real-world. Furthermore, we have added numerical studies on the impact of these approximations in Appendix C. These results suggest that our method will be of much value to our community.
> >
> > ####### Clarity issues #######
> >
> > Thanks for your suggestion. We revised the paper thoroughly to clarify some ideas that may confuse the potential audience.
> >
> > ####### References #######
> >
> > [Chen et al., 2018]: Changyou Chen, Ruiyi Zhang, Wenlin Wang, Bai Li, and Liqun Chen. A unified particleoptimization framework for scalable bayesian sampling. arXiv preprint arXiv:1805.11659, 2018.
> > [Ambrosio et al., 2008]: Luigi Ambrosio, Nicola Gigli, and Giuseppe Savare ́. Gradient flows: in metric spaces and in the space of probability measures. Springer Science & Business Media, 2008.

---

> > > ### Comment · AnonReviewer3 · 2018-11-27
> > > **Response to the revision**
> > >
> > > Thank you for the detailed response. The clarity of the paper has been improved significantly and all my questions have been reasonably addressed. I have upgraded my rating accordingly.

---

> > > > ### Author Response · Authors · 2018-12-03
> > > > **Thanks!**
> > > >
> > > > Thanks for acknowledging our contribution and revising the rating.

---

### Official Review · AnonReviewer1 · 2018-11-01
**Function Space Particle Optimization for Bayesian Neural Networks**

**Rating:** 7
**Confidence:** 3

**Review:**

This paper considers particle optimization variational inference methods for Bayesian neural networks.  To avoid degeneracies which arise when these algorithms are applied to the weight space posterior, the authors consider applying the approach in the function space.  A heuristic motivation is given for their algorithm and it seems to have good empirical performance.

I find the paper well-motivated and the suggested algorithm original and interesting.  As the authors mention at one point the derivation is rather heuristic, so much depends on the empirical assessment of their approach.  I was wondering if it was worthwhile to include an architecture search of some kind in the empirical comparisons in the examples?  This is because if wider than needed hidden layers are used this will worsen some of the degeneracies of the weight space posterior which could make the weight space algorithms perform worse.  Also the authors use a Gaussian process approximation in part of their algorithm and wide hidden layers make that approximation more reasonable and may advantage their approach for that reason too.  The authors discuss in Appendix B other approaches to improving weight space POVI.  I wonder also if parameter constraints would be helpful for improving the performance of the weight space methods, such as order constraints on the hidden layer biases for example to remove at least some of the sources of unidentifiability.  The authors talk in the introduction about the difficulties of exploring a complex high-dimensional posterior, the curse of dimensionality, and the limitations of current variational families but only 20 points are used to represent the posterior in the examples.   Are many more particles required to obtain good performance in more complex models and does the approach scale well in terms of its computational requirements in that sense?

---

> ### Author Response · Authors · 2018-11-22
> **Thanks for your feedback; response to individual questions below**
>
> Thank you for your positive and constructive feedback, and we have incorporated them into the revision. We address individual questions below.
>
> Q1: On the need of architectural search:
> A1: We address this concern from three aspects, as detailed below.
> (1) It is true that the degeneracy of weight-space POVI worsens as architecture complexity increases. We presented such an evaluation in Fig. 5, Appendix A.1. In the revision, we also added experiments on some UCI datasets using narrower network architectures in Appendix A.2.4, in which the performance of weight-space POVI methods improves, but is still outperformed by function-space methods.
> (2) We note that our previous experiment setups are standard as in BNN literature, and are fair to all baselines: in comparisons to almost all baselines, we followed the setup in their original paper. The only exceptions are the synthetic data experiments in Appendix A.1, which explicitly evaluate the impact of model complexity; and the ResNet-32 experiment, which, to our knowledge, is not considered by any previous work in BNN inference.
> (3) Finally, to model high-dimensional datasets, it is necessary to use large networks. For example, ResNet-32 has 0.4 million parameters; and the RNNs used in language modeling has hundreds of hidden units in each layer. Complex models lead to more severe over-parameterization, and BNN benchmarks should reflect this property. While we agree on the importance of evaluating different BNN architectures, e.g., for regression tasks, it requires a huge amount of work, given the number of baselines we considered, and such an evaluation should be a separate effort.
>
> Q2: On the possibility of introducing parameter constraints to improve weight space VI:
> A2: We agree that this is a good idea. However, it is hard to implement in practice, because of two reasons: 1) First, it is hard to cover all sources of unidentifiability by imposing parameter constraints. As an example, we show that order constraints alone cannot ensure identifiability: suppose we are to learn the ReLU function with a single hidden layer and a single hidden unit. We can scale the weights of the two connections in the network appropriately, and obtain an infinite number of modes; 2) Second, it is often non-trivial to apply gradient-based optimization under such constraints, and even if such an optimization scheme is possible, its impact on the learning dynamics (of variational parameters) is not clear.
> In contrast, our method eliminates all sources of unidentifiability, has a simple form similar to (unconstrained) gradient descent, and converges robustly in practice. We revised the paper to include further discussions on this, in Remark 3.1 and 3.2 where we show the resemblence of our algorithm to gradient descent; and in Appendix A.2.4 where we presented empirical evidence that our algorithm converges robustly in practice. In conclusion, we believe our method is a more practical solution to unidentifiability.
>
> Q3: On the number of particles needed, and scalability:
> A3: We address this concern from three aspects, as detailed below:
> (1) We chose to use 20 particles following the original SVGD paper, so the results are comparable; and as we have discussed in the text (Remark 3.2), our method is far more particle-efficient compared to the weight-space POVI methods.
> (2) We hypothesize that even with a small number of particles, function-space methods could produce posterior approximations that are useful in practice. A reason is that the structure of “function-space posterior” (at least in the finite-dimensional case, in which its density is well-defined) is often strikingly simple: for GP regression with conjugate likelihood, for example, the posterior is essentially uni-modal (Rasmussen, 2004). Empirically this hypothesis is supported by our experiments on CIFAR-10, in which a posterior approximation using merely 8 particles is shown to be more robust against adversarial examples, which is presumably due to the improved representation of epistemic uncertainty.
> (3) Finally, in terms of scalability, our method can be easily parallelized using model parallelism, as the only communication needed in each iteration is to broadcast the top-layer activations (function evaluations on the mini-batch). This cost is negligible compared to sending all network weights, which is needed in data-parallel training and weight-space POVI methods. We added the discussion on scalability in Appendix D.
>
> References:
> [Rasmussen, 2004]: Carl Edward Rasmussen. Gaussian processes in machine learning. In Advanced lectures on machine learning, pp. 63–71. Springer, 2004.

---

### Official Review · AnonReviewer2 · 2018-11-02
**Very promising technique, but requires clarification**

**Rating:** 7
**Confidence:** 4

**Review:**

Based on the revision, I am willing to raise the score from 5 to 7.

==========================================

The authors address the problems of variational inference in over-parameterized models and the problem of the collapse of particle-optimization-based variational inference methods (POVI). The authors propose to solve these problems by performing POVI in the space of functions instead of the weight space and propose a heuristic approximation to POVI in function spaces.

Pros:
1) I believe that this work is of great importance to the Bayesian deep learning community, and may cause a paradigm shift in this area.
2) The method performs well in practice, and alleviates the over-parameterization problem, as shown in Appendix A.
3) It seems scalable and easy to implement (and is similar to SVGD in this regard), however, some necessary details are omitted.

Cons:
1) The paper is structured nicely, but the central part of the paper, Section 3, is written poorly; many necessary details are omitted.
2) The use of proposed approximations is not justified

In order to be able to perform POVI in function-space, the authors use 4 different approximations in succession. The authors do not check the impact of those approximations empirically, and only assess the performance of the final procedure. I believe it would be beneficial to see the impact of those approximations on simple toy tasks where function-space POVI can be performed directly. Only two approximations are well-motivated (mini-batching and approximation of the prior distribution), whereas the translation of the function-space update and the choice of mu (the distribution, from which we sample mini-batches) are stated without any details.

Major concerns:
1) As far as I understand, one can see the translation of the function-space update to the weight-space update (2) as one step of SGD for the minimization of the MSE \sum_x (f(x; \theta^i) - f^i_l(x) - \eps v(f^i_l)(x))^2, where the sum is taken over the whole space X if it is finite, or over the current mini-batch otherwise. The learning rate of such update is fixed at 1. This should be clearly stated in the paper, as for now the update (2) is given without any explanation.

2) I am concerned with the theoretical justification paragraph for the update rule (3) (mini-batching). It is clear that if each marginal is matched exactly, the full posterior is also exactly matched. However, it would usually not be possible to match all marginals using parametric approximations for f(x). Moreover, it is not clear why would updates (3) even converge at all or converge to the desired point, as it is essentially the update for an optimization problem (minimization of the MSE done by SGD with a fixed learning rate), nested into a simulation problem (function-space POVI). This paragraph provides a nice intuition to why the procedure works, but theoretical justification would require more rigor.

3) Another approximation that is left unnoted is the choice of mu (the distribution over mini-batches). It seems to me from the definition of function-space POVI that we need to use the uniform distribution over the whole object space X (or, if we do not do mini-batching, we need to use the full space X). However, the choice of X seems arbitrary. For example, for MNIST data we may consider all real-values 28x28 matrices, where all elements lie on the segment [0,1]. Or, we could use the full space R^28x28. Or, we could use only the support of the empirical distribution. I have several concerns here:
3.1) If the particles are parametric, the solution may greatly depend on the choice of X. As the empirical distribution has a finite support, it would be dominated by other points unless the data points are reweighted. And as the likelihood does not depend on the out-of-dataset samples, all particles f^i would collapse into prior, completely ignoring the training data.
3.2) If the prior is non-parametric, f(x) for all out-of-dataset objects x would collapse to the prior, whereas the f(x) for all the training objects would perfectly match the training data. Therefore we would not be able to make non-trivial predictions for the objects that are not contained in the training set unless the function-space kernel of the function-space prior somehow prevents it. This poses a question: how can we ensure the ability of our particles to interpolate and extrapolate without making them parametric? Even in the parametric case, if we have no additional regularization and flexible enough models, they could overfit and have a similar problem.
These two concerns may be wrong, as I did not fully understand how the function-space prior distribution works, and how the function-space kernel is defined (see concern 4).

4) Finally, it is not stated how the kernels for function-space POVI are defined. Therefore, it is not clear how to implement the proposed technique, and how to reproduce the results. Also, without the full expression for the weight-space update, it is difficult to relate the proposed procedure to the plain weight-space POVI with the function value kernel, discussed in Appendix B.

Minor comments:
1) It is hard to see the initial accuracy of different models from Figure 3 (accuracy without adversarial examples). Also, what is the test log-likelihood of these models?
2) It seems that sign in line 5 on page 4 should be '-'

I believe that this could be a very strong paper. Unfortunately, the paper lacks a lot of important details, and I do not think that it is ready for publication in its current form.

---

> ### Author Response · Authors · 2018-11-22
> **Thanks for your review; response and clarification (1/2)**
>
> Thank you for your positive and constructive feedbacks. We revised the presentation thoroughly following them. However, there are also some misunderstandings that we hope to clarify.
>
> Q: On lack of details:
> A: We clarified the kernel specification issue you mentioned in Section 3.1.2. Our implementation will also be made public after the review process to make everything reproducible.
>
> Q: On the justification of approximations:
> A: Thanks for the suggestions. First, as you suggested, we have added simulations evaluating the impact of the parametric approximation and mini-batching in Appendix C. The result shows they do not influence convergence. Second, we have thoroughly revised the justification for stochastic approximation in Section 3.1.2.
> Finally, please note that your comment on the choice of \mu being an approximation could be a major understanding, which we clarify below.
>
> Regarding your major concerns:
>
> Q1. Regarding the weight space update (2) (now Eq.(3)):
> A1: Thank you for the comments. Yes, your explanation is correct. We have revised Section 3.1.1 to clarify it. In brief, such an update is easy to implement, relates to ensemble training, and does not impact convergence empirically. Please read the revision for full details.
>
> Q2. Regarding the theoretical justification of (3) (now Eq.(4)):
> A2: Thanks for the comments. We clarify from three points:
> (1) On your concern that marginals could not be matched exactly: we added a paragraph in the `Justificiation of (4)` part in Sec 3.1.2. The point is that even in that case, the average energy is an excellent choice of variational objective, as in practice we only care about average approximation error of lower-order moments; and in the definition of the averaged energy, \mu could be specified to incorporate distributional assumptions about the test set.
> (2) On convergence: in Appendix C, we added synthetic experiments specifically verifying that parametric approximation plus mini-batching does not impact convergence. We also added a convergence plot for a real-world dataset in Appendix A.2.4, which shows our algorithm is fairly stable in practice.
> (3) However, a rigorous theory that simultaneously addresses both approximations could be hard to develop, as our procedure works on the infinite-dimensional Wasserstein spaces, a Riemannian manifold with a non-Euclidean metric; and optimization (i.e. simulation of gradient flows) on Riemannian manifolds is less well-studied than their Euclidean counterparts. We are not aware of any immediate results to this answer.
> Although a bit heuristic, our method is well-motivated, works well empirically, and has an elegant, easy-to-implement form. We hope you agree that such a procedure could be of value to the community.
>
> Q3. On your belief that we need a uniform \mu:
> A3: There is some misunderstanding that might be caused by previous typos and over-restrictive conditions in Section 3.1.2. We apologize for the confusion; the text has been revised, and we also clarify this issue below. In fact, the only requirement on \mu is that
>
>   if q(f(x))=p(f(x)|X_{train},Y_{train}) almost everywhere w.r.t. \mu(x), then q(f) and p(f|X_{train},Y_{train}) defines the same stochastic process.
>
> For example, if the posterior is a GP, a sufficient condition is that B>=2, and a single sample from \mu consists of samples from a continuous measure supported on the entire X, as well as samples from the training set. According to the condition above, \mu don’t need to have identically distributed components (recall it is a distribution on X^B, and a sample from \mu contains B samples in the input space), or follow the uniform distribution.
> As mentioned in the second to last paragraph in Section 3.1.2, our choice of \mu is the product measure of mini-batch from the training set, and samples from its kernel density estimation.
>
> Q3.1. On your concern that training set could be ignored:
> A3.1: As clarified above in our response to Q3, a single sample from \mu contains both training data and samples from a continuous distribution, so training data is not ignored.

---

> > ### Author Response · Authors · 2018-11-22
> > **Response and Clarification (2/2)**
> >
> > Q3.2-i: On your concern that f(x) could collapse to prior:
> > A3.2-i: You are correct that the function-space prior prevents it: while priors used in practice (like the common weak Gaussian prior on NN weights) are flexible, their flexibility is far from letting out-of-sample prediction collapse to the prior, as they all encode smoothness in some sense (See the example in [^1] below). The particle functions in your example are a.s. not continuous, and are not inside the support of the prior. As the support of posterior must be the subset of that of prior, the situation your described *will not happen*. You can also see it from the synthetic experiments in Section 5.1.
> > [^1]: E.g. for GP with a RBF kernel, samples from prior are a.s. continuous; the BNN with truncated priors only represents functions with a bounded Lipschitz constant. (It is possible to do inference using non-smooth priors, but they must also encode correlation, and can be analyzed similarly. It seems that the only prior satisfying your description is an i.i.d. noise process, which should not be used in practice anyway.)
> >
> > Q3.2-ii: A further clarification on overfitting in parametric models:
> > A3.2-ii: (1) It is still possible that with a pathological prior (or one that is too weak), prediction based on full posterior still overfits (See this blog article and references therein: http://www.nowozin.net/sebastian/blog/do-bayesians-overfit.html). However, this is not a problem that can be solved by the inference side, but rather the problem of specifying right priors. In other words, if the prior is pathological, a faithful inference procedure should make the user aware of that.
> > (2) Also, even for imperfect priors, Bayesian inference guards against overfitting much better than the MAP estimate, and the observation that frequentist inference for NN overfits does not directly transfer to the Bayesian case. An intuitive explanation is that Bayesian inference performs model averaging, which weights each model point based on its complexity.
> >
> > Q4. On the definition of kernels in function-space POVI, and our method’s relation to the function value kernel:
> > A6: (1) We apologize for the mistake; we added it back to Section 3.1.2. Please notice that gradient flows are defined on B-dimensional marginal distributions, thus kernels in our algorithm are defined on finite-dimensional spaces, and can be easily specified.
> > (2) As for the relation between our proposal and the function-value kernel, we have added a new Appendix D, in which we derived the detailed update rules and compared them. We hope it clarifies your concerns.
> >
> > Finally, for your minor comments, we address them as detailed below:
> >
> > 1. For the adversarial defense experiment, we have added the initial accuracy and log likelihood in Appendix A.3.
> > 2. We have fixed the typo on Page 5, line 4. Thanks for pointing it out.

---

> > > ### Comment · AnonReviewer2 · 2018-11-26
> > > **Response to the revision**
> > >
> > > Thank you for clarification; the paper has become much more clear after the revision, and the extended experiments provide additional motivation/justification for the proposed approximations. Most of my concerns have been answered.
> > >
> > > I still have several questions and comments:
> > >
> > > 1) I am still convinced that the predictive performance of the model would greatly depend on the choice of mu, although this choice *seems* to be arbitrary. However, now I see it differently: as we are using the parametric approximation, it is not possible to perfectly approximate the posterior on the full set of X. By choosing different distributions mu, we can choose the regions of the input space in which we would like to approximate the posterior better. For example, if we seek better predictive performance, it makes perfect sense to use the KDE of the training data and use unlabeled data to gather more information about the domain. If one is interested in good out-of-domain uncertainty, it would make sense to add some out-of-domain data. I believe that the careful choice of mu is a nice way to incorporate the domain information in the trained discriminative model.
> > >
> > > 2) How would one balance B' and B-B'? How does it impact the uncertainty and predictive performance? What values of B and B' do you use in the experiments?
> > >
> > > 3) What do you mean by "f(x) is already high-dimensional for a single data point x, one should down-sample f(x) within each x"? If I understand correctly, dim f(x) is equal to the number of classes, which is typically not very high. Could you elaborate more on the mentioned down-sampling? How exactly is it performed, and why is it needed?
> > >
> > > 4) I am not sure whether the accuracy degradation in Table 10 can be attributed to adversarial robustness. If I understand correctly, you do not perform adversarial training or otherwise address adversarial examples explicitly. Still, it would be interesting to see how the choice of B' and the parameters of KDE would influence the prediction performance and the robustness to adversarial examples. I suspect that it is possible to obtain some trade-off.
> > >
> > > To sum up:
> > > + The paper provides an elegant way to perform function-space posterior inference that does not suffer from overparameterization
> > > + The paper provides a nice way to explicitly choose the regions in the input space in which the posterior should be approximated better
> > > - Although the approximations are well-motivated, there are many of them. It is not clear how well does the proposed procedure correspond to the true posterior inference, or how different parameters of the procedure impact the inference.

---

> > > > ### Author Response · Authors · 2018-12-03
> > > > **Thanks & Response**
> > > >
> > > > Thank you for acknowledging our contributions, revising the ratings and the nice comments. We appreciate that. Below, we briefly address the extra questions.
> > > >
> > > > Q1: Regarding the impact of B' on performance, and values we have used in the experiments
> > > > A1: We will make them clearer in the final version. In fact, (1) we did not tune B' thoroughly, and only experimented with B'<=B/2; inside this range, increasing B' improves predictive performance, although improvement becomes marginal when B' is large. E.g. on the Concrete dataset in Section 5.2.1, setting B'=100 improves the average NLL by 0.04 compared to B'=10, and by 0.08 compared to B'=0; the standard deviation of NLL on this dataset is 0.04.
> > > > (2) In the synthetic experiment in Section 5.1, varying B' does not have a significant impact on the quality of uncertainty estimation. A possible explanation is that the smoothness constraint encoded in the function-space prior "propagates" uncertainty in q[f(X_{train})] to q[f(x)] for nearby x.
> > > > (3) The value of B-B' is given in the text. For all experiments on feed-forward networks, we have set B' to min(100, B/2). This value is determined by grid search (in {1, 10, 100}) on a UCI regression dataset. In the ResNet experiment we used B'=4, and we did not experiment with other values.
> > > >
> > > > Q2: On downsampling f(x) in classification
> > > > A2: We apologize that the phrase "for a single data point x" might be misleading here. We will make it clearer in the final version. In fact, denote the batch size (used for prior estimation) as B and number of classes as C, then without down-sampling, we will need to approximate the prior distribution of a B*C dimensional vector, the concatenated function values for all data points in the batch. This is high-dimensional compared to the 1d-regression case, where the dimension of the concatenated function values is B.
> > > > So we choose to down-sample the indices of this vector, i.e. to down-sample the set of classes for which we will take the corresponding logits, concatenate them for all data points in a batch, and estimate the prior distribution. It is a type of stochastic approximation similar to mini-batching, and is not necessary: alternatively we can use a smaller B.
> > > >
> > > > Q3-i: On the accuracy on clean data in the adversarial robustness experiment:
> > > > A3-i: We will clarify this in more detail in the final version. As widely observed in the literature (e.g., Liao et al., 2017), it is common to sacrifice some (often tiny) performance on clean samples in order to obtain significant improvement in adversarial robustness. For BNNs, under certain priors, the posterior mean estimate could be sub-optimal on iid test samples, compared to an ensemble of MAP estimates; and it is possible the prior we used in the ResNet experiment falls into this case. However, such a prior can still be useful for adversarial defense, because, as we have reviewed in Section 5.3, the latter task involves prediction on uncertain inputs. As we focused on adversarial robustness, we did not adjust the prior specifically to optimize accuracy on clean data, and used the Gaussian prior corresponding to the original L2 regularizer in ResNet instead. We leave the search of a more sensible prior as future work.
> > > >
> > > > Q3-ii: On the possibility to adjust \mu for robustness applications:
> > > > A3-ii: Thanks for the suggestion. We agree that it will be an interesting direction of future work. E.g. we could add to \mu a component that focuses on the regions of adversarial examples, similar to what we proposed for domain adaptation; and to improve performance on general CV tasks, in KDE we could specify better kernels than isotropic Gaussian kernels. It is possible that a careful adaptation of our method could further improve adversarial robustness, or performance on general CV tasks.

---

### Author Response · Authors · 2018-11-22
**Summary of the Revision**

- We revised Section 3 thoroughly. Please notice the change of equation numbering.
- We added three remarks in Section 3.1.1, discussing the motivation of the parametric update rule (2) (i.e., Eq (3) in the revision); comparing it with weight-space POVI and ensemble training; and noting that it does not impact convergence in synthetic experiments.
- We revised Section 3.1.2 to clarify a few possible misunderstandings, including the requirements of the sampling distribution \mu and the specification of the kernel.
- In Appendix A (experimental details and further results), we added convergence plots and benchmark for narrower network architectures on UCI datasets in Section A.2.4; and accuracy and log likelihood on clean MNIST and CIFAR-10 data in Section A.3.
- We added a new appendix, Appendix C, where we evaluated the impact of several approximations in our algorithm.
- We added a new appendix, Appendix D, where we further clarified the connection and differences between our algorithm, weight-space POVI, and ensemble methods.
- We revised the language and fixed a few typos, most notably:
    - In Algorithm 1, we fixed a typo where previously, we erroneously required to sample from \mu *and* a batch from the training set. In the corrected version, the training-set batch is *part of* samples from \mu.
    - In Appendix A.2.3, we fixed the results of the “protein” dataset in Table 7 (comparison with Ma et al. (2018)). The previous result was obtained using an incorrect configuration, due to a scripting error. We apologize for the inconvenience; however, *neither the comparison result on that specific dataset, or the conclusion of the corresponding experiment is influenced*.

---

### Public Comment · ~Roman_Novak2 · 2019-01-11
**On function-space priors for wide BNNs**

Dear Colleagues,

Congratulations on having your impressive work accepted!

We similarly consider BNNs to be a very important and exciting field of study, and investigate them in the wide regime, where equivalence to the GP prior arises. As you reference this line of work in section 3.2, we would like to point out relevant concurrent work:

1) https://openreview.net/forum?id=B1EA-M-0Z derived the NN-GP correspondence concurrently with Matthews et al, 2018, and
2) https://openreview.net/forum?id=B1g30j0qF7 similarly derived the CNN-GP correspondence concurrently with Garriga-Alonso et al, 2018.

In addition to being concurrent derivations, both works provide a lot of complementary findings. I hope you find these references useful!

Best,
Roman.

---

> ### Author Response · Authors · 2019-01-12
> **Thanks for the pointers**
>
> Hi Roman,
>
> Thanks for the pointers! I just went through these papers and, indeed, it was an enjoyable read. We will add citations in the camera-ready version.
>
> Best,
> Ziyu Wang

---

### Meta-Review · Area_Chair1 · 2018-12-14

**Confidence:** 4
**Recommendation:** Accept (Poster)

**Metareview:**

Reviewers are in a consensus and recommended to accept after engaging with the authors. Please take reviewers' comments into consideration to improve your submission for the camera ready.